# AGENT-CHAINED POLICY OPTIMIZATION

## ABSTRACT

We study Cooperative Multi-Agent Reinforcement Learning (MARL), where the aim is to train decentralized policies that maximize a shared return. Existing methods typically employ either iterative best-response updates, which converge only to Nash Equilibria (NE) that may be far from the global optimum, or simultaneous learning with centralized critics, which lack convergence guarantees to the optimal joint policy without strong assumptions on decomposable value functions. We introduce the Agent-Chained Belief MDP (AC-BMDP), which reformulates MARL as a serialized decision process where agents act sequentially while maintaining beliefs over actions taken by preceding agents. This enables the definition of agent-specific value functions that are naturally chained together. Building on this framework, we propose Agent-Chained Policy Iteration (ACPI) and prove that it converges to the globally optimal joint policy. We further develop this framework into a practical actor–critic algorithm, Agent-Chained Policy Optimization (ACPO). On standard benchmarks, ACPO consistently surpasses state-of-the-art baselines, with the performance advantage growing significantly as the number of agents increases.

## 1 INTRODUCTION

Cooperative multi-agent systems are increasingly employed to tackle complex tasks ranging from autonomous vehicle fleets to distributed sensor networks, where decentralized coordination is required (Dafoe et al., 2020). The goal of Cooperative Multi-Agent Reinforcement Learning (MARL) is to learn a set of policies that enable multiple agents to collectively maximize a shared return. Despite substantial progress, achieving decentralized (fully factorized) policies that provably converge to the global optimum remains an open challenge.

Existing MARL methods largely fall into one of two categories: iterative best-response updates or simultaneous policy updates. First, iterative best-response updates bypass the problem of joint policy optimization by instead solving for a game-theoretic equilibrium. For instance, Multi-Agent Policy Iteration (MA-PI) (Zhong et al., 2024) adopts the iterative best-response approach which is designed to converge to a Nash equilibrium (NE), and provides the foundation for practical algorithms such as HAPPO (Kuba et al., 2022) and HATRPO (Zhong et al., 2024). However, NEs are a fixed point where policies cannot be unilaterally improved, and this can be arbitrarily far from the optimal joint policy in fully cooperative settings(see Table 1 for a simple illustration). Furthermore, they require each agent to be updated in turn while holding the other agents fixed. When these methods are extended to high-dimensional settings, this prolongs training time and prohibits scaling to a larger number of agents in practice.

The second approach is simultaneous policy updates, which considers the multi-agent problem as a single-agent MDP defined over the joint action space. A wide range of algorithms follow this paradigm, which includes, but not limited to, MAPPO (Yu et al., 2022) and MADDPG (Lowe et al., 2017). These methods are natural extensions of single-agent RL, where all agents are trained in parallel, making them simple to implement and scalable in practice. However, they typically rely on a centralized value function defined over the joint action space, while policies must remain decentralized during execution. This mismatch creates a gap between training and deployment, and convergence to an optimal policy often requires additional assumptions such as value decomposition (Rashid et al., 2018; Zhang et al., 2021).

Figure 1: Centralized Critics (left) commonly used in prior work and Decentralized Critics (right) with agent chaining used in ACPO.

In summary, state-of-the-art MARL methods can largely be categorized into iterative best-response updates or simultaneous updates. Iterative best-response methods offer convergence guarantees, but only to an NE, and their sequential updates lead to prohibitive training costs as the number of agents increases. Simultaneous update methods avoid this bottleneck but lack convergence guarantees unless strong assumptions are imposed. Ultimately, neither class is fundamentally designed to ensure convergence to the globally optimal joint policy that maximizes return.

In this work, we leverage a key property of MARL: Any simultaneous problem can be recast as a sequential one where the agents take actions sequentially. However, this serialized version of the problem cannot be solved directly since each agent cannot observe the actions executed by other agents. We formalize this by introducing the Agent-Chained Belief MDP (AC-BMDP), which is an MDP with the belief over the actions by preceding agents. While the formulation chains agents sequentially, each agent still executes its own policy independently, consistent with the CTDE paradigm. The combination of serialization and the notion of a belief creates well-defined agent-specific value functions which are naturally *chained* to the value function of the next agent.

Our main theoretical contribution is Agent-Chained Policy Iteration (ACPI), which we prove convergence to the globally optimal joint policy in the underlying Multi-Agent MDP. Building on ACPI, we introduce Agent-Chained Policy Optimization (ACPO), a practical actor-critic algorithm under the Centralized Training with Decentralized Execution (CTDE) setting. Unlike iterative best-response updates, our formulation aims for optimal cooperative policies and facilitates parallel policy updates. Also, unlike simultaneous policy updates where the centralized value functions and fully factorized policies are not aligned without proper assumptions, our agent-chained formulation ensures that value functions and policies are inherently aligned (see Figure 1).

Empirically, we evaluate our approach on a suite of standard MARL benchmarks, including Multi-Robot Warehouse, SMACv2 and MA-MuJoCo. We show that ACPO consistently outperforms strong baselines on all tasks, where the gap widens as the number of agents increases. To the best of our knowledge, ACPO is the first algorithm under CTDE that directly targets convergence to the globally optimal policy, instead of a Nash equilibrium.

## 2 BACKGROUND

### 2.1 MULTI-AGENT MDP

We consider a cooperative multi-agent environment with $\mathcal{N} = \{1, \dots, N\}$ agents, formally defined as a Multi-Agent Markov Decision Process (MMDP) (Boutilier, 1996). At time step $t$, each agent $i \in \mathcal{N}$ takes action $a_t^{(i)} \in A^{(i)}$ sampled from policy $\pi^{(i)}(a_t^{(i)} \mid s_t)$ where $s_t \in S$ is the state. The state transition is Markovian, i.e. the next state $s_{t+1}$ is given by transition function $T(s_{t+1}|s_t, \vec{a}_t)$ where $\vec{a}_t$ is the joint action $\vec{a}_t = [a_t^{(1)}, \dots, a_t^{(N)}]$. Each agent receives shared reward $r_t$ generated by the common reward function $R(s_t, \vec{a}_t)$.

The goal of the multi-agent RL task is to find a set of agent policies $\vec{\pi} = [\pi^{(1)}, \dots, \pi^{(N)}]$ that maximize the total expected return $J = \mathbb{E}_{\tau \sim \Pr(\tau|\vec{\pi})}[\sum_t \gamma^t r_t]$ where $\gamma$ is the discount factor. As in previous work on cooperative MARL, we assume that the policies $\vec{\pi}$ are allowed to be trained in a centralized manner, but require decentralized execution, known as the Centralized Training with Decentralized Execution (CTDE) setting.

## 2.2 PREVIOUS APPROACHES IN COOPERATIVE MARL

**Iterative best-response update (Iterative BR)** One way to solve MMDPs is to define a reduced MDP for each agent and learn the best response policy in order to converge to a Nash equilibrium (NE) (Bertsekas, 2020). Formally, each agent $i$ solves its own reduced MDP $\langle S, A^{(i)}, T^{(i)}, R^{(i)} \rangle$, where $T^{(i)}(s_{t+1}|s_t, a_t^{(i)}) := \mathbb{E}_{\vec{\pi}^{-i}}[T(s_{t+1}|s_t, a_t^{(i)}, \vec{a}^{-i})]$, and $\vec{a}^{-i}$ is the joint action excluding the action of agent $i$ and is distributed according to the most recent policies $\vec{\pi}^{-i}$. The reward function is marginalized similarly as $R_t^{(i)}(s_t, a_t^{(i)}) = \mathbb{E}_{\vec{\pi}^{-i}}[R(s_t, a_t^{(i)}, \vec{a}^{-i})]$. Recent methods such as HATRPO (Kuba et al., 2022) and HAPPO (Zhong et al., 2024) leverage this insight and use single-agent RL methods such as TRPO (Schulman et al., 2015) and PPO (Schulman et al., 2017) to train the best response policy for each agent, respectively. However, this only approximates to an NE where each agent cannot unilaterally improve its policy, or Quantal Response Equilibrium (QRE) (Liu et al., 2024) which is an NE defined over the entropy-augmented reward function. Crucially, there is no guarantee that this NE or QRE will coincide with a globally optimal policy in the underlying MMDP. Furthermore, Iterative BR does not scale well in practice. Since each agent is trained sequentially while holding the other agent fixed, training time significantly increases as the number of agents increase.

**Simultaneous policy update** An alternative way to solve MMDPs is to simply view the problem as a single-agent MDP defined over the joint action space. Previous methods such as MAPPO (Yu et al., 2022) and MADDPG (Lowe et al., 2017) train a centralized critic over this MDP. MAPPO, for instance, trains each policy via an objective of the form, $J(\pi^{(i)}) = \mathbb{E}_{\pi^{(i)}, \vec{\pi}^{-i}} \left[ A(s, a^{(i)}, \vec{a}^{-i}) \right]$ where $A(s, \vec{a})$ is the advantage computed via Generalized Advantage Estimation (GAE) (Schulman et al., 2016). At first glance, it may seem that this approach is maximizing return over factorized policies, since each policy $\pi^{(i)}$ is trained simultaneously. However, a more careful look reveals that each agent updates its policy to maximize $A^{(i)}(s, a^{(i)})$ where $\vec{a}^{-i}$ is marginalized out. This does not guarantee that the joint policy will be improved unless we impose strong assumptions such as value decomposition (Rashid et al., 2018; Zhang et al., 2021). In general, the convergence guarantees of these methods are not well-understood. In comparison to Iterative BR approaches, the parallel nature of the policy updates mitigates the bottleneck during training time.

## 2.3 LIMITATION OF PREVIOUS APPROACHES

Consider the simple Matrix Game in Table 1 with 2 agents and 3 actions, which was considered in Liu et al. (2024). The three NEs are $(A, A), (B, B), (C, C)$, since there is no incentive for either agent to change its action at each of those NEs. $(C, C)$ is a global optimum of the game since it achieves the highest return. As shown in Table 1, Iterative BR and simultaneous policy update, which are the foundations for HAPPO and MAPPO, respectively, cannot escape suboptimal NEs. The only way it can converge to $(C, C)$ is for the policy to be initialized with high probability towards $(C, C)$. However, it is not desirable to rely on the initialization of the policy for finding good NEs, since it implies that extensive hyperparameter tuning is required. Similarly, Heterogenous-Agent Soft Policy Iteration (HASPI) (Liu et al., 2024) can only find the global optimum for specific combinations of the entropy parameter $\alpha$ and the initialization of the policy (e.g. $\alpha = 5$ and $\pi^{(1)}(A) = \pi^{(2)}(A) = 0.6$), if it happens to coincide with the QRE. Similar to NE, there is no guarantee that the QRE will coincide with the global optimum, and thus generally requires extensive hyperparameter tuning to find the right balance between reward and entropy maximization.

Our goal in the rest of the paper is to derive a principled algorithm which directly targets the globally optimal policy in the MMDP. In Section 5, we present Agent-Chained Policy Iteration (ACPI) which converges to the global optimum regardless of the policy initialization. Detailed analysis on how ACPI is able to solve the Matrix Game introduced in Table 1 is provided in Appendix G.

## 3 SERIALIZATION

To address the challenges in simultaneous decision-making by multiple agents, we adopt a key property of MMDPs, namely that we can solve an equivalent serialized version of the problem, where each agent takes actions sequentially. This serialization approach, while explored in Bertsekas (2019), Kovařík et al. (2023) and Peralez et al. (2025) has not been successfully utilized in the Deep

|     | $A$ | $B$ | $C$ |
| --- | --- | --- | --- |
| $A$ | 5 | $-20$ | $-20$ |
| $B$ | $-20$ | 10 | $-20$ |
| $C$ | $-20$ | $-20$ | 20 |

3x3 Matrix Game

| | $A$ | $B$ | $C$ | | $A$ | $B$ | $C$ | | $A$ | $B$ | $C$ | | $A$ | $B$ | $C$ | | $A$ | $B$ | $C$ |
| --- | --- | --- | --- | --- | --- | --- | --- | --- | --- | --- | --- | --- | --- | --- | --- | --- | --- | --- | --- |
| $A$ | 1 | 0 | 0 | | 1 | 0 | 0 | | 1 | 0 | 0 | | 0 | 0 | 0 | | 0 | 0 | 0 |
| $B$ | 0 | 0 | 0 | | 0 | 0 | 0 | | 0 | 0 | 0 | | 0 | 0 | 0 | | 0 | 0 | 0 |
| $C$ | 0 | 0 | 0 | | 0 | 0 | 0 | | 0 | 0 | 0 | | 0 | 0 | 1 | | 0 | 0 | 1 |
| | | Simul. Update | | | | Iterative BR | | | | HASPI ($\alpha$=1) | | | | HASPI ($\alpha$=5) | | | | ACPI | |

Table 1: A 3x3 Matrix Game and converged policy values when initialized to $\pi^{(1)}(A) = \pi^{(2)}(A) = 0.6$. ACPI (Ours) converges to the global optimum $(C, C)$ regardless of the initialization.

MARL setting. Specifically, each time step $t$ is decomposed into $N$ micro-steps. Thus, there are now two timescales at each decision point: the time step denoted by $t$ and micro-steps denoted by $t'$, where $\lfloor t'/N \rfloor = t$. Each micro-step $t'$ corresponds to an agent $i$ committing to individually executing action $a^{(i)}$. Under this framework, the MMDP state is now augmented to include the actions taken by previous agents, i.e. $[s_t, \vec{a}_t^{<i}]$, where $\vec{a}_t^{<i} = [a_t^{(1)}, \ldots, a_t^{(i-1)}]$. The state transitions within micro-steps are deterministic. A full state transition by $T$ only occurs after all $N$ agents have committed their actions:

$$T([s_t, \vec{a}_t^{<i+1}] \mid [s_t, \vec{a}_t^{<i}], a_t^{(i)}) = \mathbb{I}\{[\vec{a}_t^{<i}, a_t^{(i)}] = \vec{a}_t^{<i+1}\} \text{ if } i \in \{1, \ldots, N-1\}$$

$$T([s_{t+1}, \emptyset] \mid [s_t, \vec{a}_t^{<N}], a_t^{(N)}) = T(s_{t+1} \mid s_t, \vec{a}_t)$$

Similarly, rewards are only generated by MMDP reward function $R$ once all the agents have committed their actions.

$$R([s_t, \vec{a}_t^{<i}], a_t^{(i)}) = \begin{cases} R(s_t, \vec{a}_t) & \text{if } i = N \\ 0 & \text{otherwise} \end{cases}$$

Finally, the discount factor is denoted as $\gamma'$ where $\gamma' = \gamma^{1/N}$.

The optimal policy for this serialized problem is in fact optimal for the original MMDP as well.

**Theorem 3.1.** *(Peralez et al., 2025) For every MMDP, there exists a serialized multi-agent problem, of which its optimal policy is also optimal for the underlying MMDP.*

Serialization has a number of practical advantages. First, by defining the action space on individual actions $|A^{(i)}|$, the action space of each timestep is reduced. It effectively mitigates the exponential complexity associated with the joint action space $|A| = |A^{(i)}|^N$ for search. Second, serialization inherently facilitates credit assignment across individual actions as all of the components of the MDP are defined with respect to individual actions. In contrast, the reward function $R$ in the original problem is a function of the joint action $\vec{a}$, and thus requires a separate credit assignment mechanism (e.g. Foerster et al. (2018); Wang et al. (2022)) to attribute contributions of individual actions $a^{(i)}$ to overall rewards and returns. Finally, serialization changes the perspective of the problem into a single-agent MDP, thus making it easier to apply single-agent techniques and algorithms.

## 4 AGENT-CHAINED BELIEF MDP (AC-BMDP)

While serialization transforms the multi-agent problem into a sequential single-agent MDP over micro-steps and individual actions, a critical challenge remains: the actions taken by other agents are not observable. *This effectively renders the serialized problem as a partially observable one, even if the underlying problem is fully observable.* As a consequence, this necessitates the definition of a belief MDP, where the belief state represents a distribution over the actions taken by other agents. To this end, we propose the Agent-Chained Belief MDP (AC-BMDP), which is designed such that (1) the optimal policy coincides with that of the underlying MMDP and (2) incorporates the notion of a belief over preceding actions.

Motivated by (Nayyar et al., 2013), The action space in an AC-BMDP is no longer the actions $a^{(i)}$, but rather an action distribution $\phi^{(i)} \in \Delta(A^{(i)})$. The belief $b_t^{(i)}(\vec{a}_t^{<i}) \triangleq \Pr(\vec{a}_t^{<i} \mid s_t, \vec{\phi}_t^{<i})$ for agents $i \in \{2, \ldots, N\}$ is defined as the distribution over the previous agent actions within the current time

step $t$. For agent 1, the belief is empty as there are no preceding agents. The state space $[s_t, b_t^{(i)}]$ now augments the state $s_t$ with agent $i$'s current belief. The reward function and transition function are defined as follows:

$$R\left([s_t, b_t^{(i)}], \phi_t^{(i)}\right) = \sum_{\vec{a}_t^{<i}} b_t^{(i)}(\vec{a}_t^{<i}) \sum_{a_t^{(i)}} \phi_t^{(i)}(a_t^{(i)}) R\left([s_t, \vec{a}_t^{<i}], a_t^{(i)}\right).$$

$$T\left([s_t, b_t^{(i+1)}] \mid [s_t, b_t^{(i)}], \phi_t^{(i)}\right) = \begin{cases} 1 & \text{if } b_t^{(i+1)} = \tau\left([s_t, b_t^{(i)}], \phi_t^{(i)}\right) \\ 0 & \text{otherwise} \end{cases} \quad \text{if } i \in \{1, \ldots, N-1\}$$

$$T([s_{t+1}, \emptyset] \mid [s_t, b_t^{(N)}], \phi_t^{(N)}) = \sum_{\vec{a}^{<N}} b_t^{(N)}(\vec{a}_t^{<N}) \sum_{a_t^{(N)}} \phi_t^{(N)}(a_t^{(N)}) T(s_{t+1} \mid s_t, \vec{a}_t^{<N}, a_t^{(N)})$$

where $\tau$ is the belief update rule provided in Appendix A.

Our formulation as an AC-BMDP highlights the fact that, by the nature of simultaneous action selection, each agent must infer the previous agents' actions in order to make optimal decisions. It is worth noting that opponent modelling and the prediction of other agents' actions have a rich history in MARL (Albrecht & Stone, 2018). However, while previous work focused on opponent modelling as an additional algorithmic module, we derive this formally as something necessary for solving the underlying MMDP and finding the globally optimal policy.

## 5 AGENT-CHAINED POLICY ITERATION

In this section, we present a policy iteration procedure called *Agent-Chained Policy Iteration* (ACPI) which is defined on the AC-BMDP, and formally prove that the fixed point of this procedure is also optimal in the underlying MMDP. Unlike Iterative BR (Zhong et al., 2024) where the fixed point of policy iteration is a NE, we show that ACPI is guaranteed to converge to the global optimum of the MMDP.

We start by defining the Bellman operator under the AC-BMDP.

**Definition 5.1.** (Agent-Chained Bellman Operators)

$$(\mathcal{T}^{\vec{\pi}} Q^{(1)})([s, b^{(N)}], \phi^{(N)}) := R\left([s, b^{(N)}], \phi^{(N)}\right) + \gamma' \mathbb{E}_{\substack{s' \sim T(\cdot \mid [s, b^{(N)}], \phi^{(N)}) \\ \phi^{(1)} \sim \pi^{(1)}(\cdot \mid s')}} \left[Q^{(1)}\left(s', \phi^{(1)}\right)\right]$$

$$(\mathcal{T}^{\vec{\pi}} Q^{(i+1)})([s, b^{(i)}], \phi^{(i)}) := \gamma' \mathbb{E}_{\substack{b^{(i+1)} = T([s, b^{(i)}], \phi^{(i)}) \\ \phi^{(i+1)} \sim \pi^{(i+1)}(\cdot \mid s, b^{(i+1)})}} \left[Q^{(i+1)}\left([s, b^{(i+1)}], \phi^{(i+1)}\right)\right]$$

$$\text{if } i \in \{1, \ldots, N-1\},$$

where we used the fact that $R = 0$ for the micro-steps when agents $i \in \{1, \ldots N-1\}$ take actions. Note that we have defined the Bellman operators separately for agent $N$ since its actions will affect the actual transitions and rewards.

We immediately see the benefit of serialization, where we now have a well-defined set of *decentralized value functions* for each agent. This is in sharp contrast to previous approaches which required restrictions on the environment to decompose the value function into individual utility functions (Peng et al., 2021; Rashid et al., 2018; Zhang et al., 2021).

The Q-values also have the intended meaning, which is the expected return given that agent $i$ is in state $[s, b^{(i)}]$ and takes action distribution $\phi^{(i)}$ (which reduces to action $a^{(i)}$ for deterministic $\phi^{(i)}$). This now provides us with a natural way to assign credit for each agent. Moreover, we have now *chained* the agents by their Q-values, as the target for $Q^{(i)}$ is $Q^{(i+1)}$. This provides an intuitive interpretation for policy evaluation, where $Q^{(i)}([s, b^{(i)}], \phi^{(i)})$ considers how taking an action $\phi^{(i)}$ at state $[s, b^{(i)}]$ will affect the next agent's Q-values, $Q^{(i+1)}$.

By repeatedly applying $\mathcal{T}^{\vec{\pi}}$, we can obtain the $Q$-values for a given joint policy $\vec{\pi}$:

**Lemma 5.2.** *(Agent-Chained Policy Evaluation) The Agent-Chained Bellman Operators in Definition 5.1 are a contraction mapping under the infinity norm. Thus, starting with any $\vec{Q} = \langle Q^{(1)}, \ldots Q^{(N)} \rangle$ and a joint policy $\vec{\pi} = \langle \pi^{(1)}, \ldots, \pi^{(N)} \rangle$, the repeated application of $\mathcal{T}^{\vec{\pi}}$ will return a set of Q-values for each agent $\langle Q^{(1, \vec{\pi})}, \ldots Q^{(N, \vec{\pi})} \rangle$ in the limit.*

*Proof.* See Appendix C.1. □

During policy improvement, each agent's policy $\pi^{(i)}$ will be updated to select the greedy action with respect to their own $Q^{(i)}$,

$$\forall i, b^{(i)}, s, \pi_{new}^{(i)}([s, b^{(i)}]) \leftarrow \arg\max_{\phi^{(i)}} Q^{(i,\vec{\pi})}([s, b^{(i)}], \phi^{(i)}). \tag{1}$$

**Lemma 5.3.** *(Agent-Chained Policy Improvement) Given a deterministic policy $\vec{\pi} = \langle \pi^{(1)}, \ldots, \pi^{(N)} \rangle$, let $Q^{(i,\vec{\pi})}$ denote the i-th agent's value function for a joint policy $\vec{\pi}$. If we update the new policy $\vec{\pi}_{new} = \langle \pi_{new}^{(1)}, \ldots, \pi_{new}^{(N)} \rangle$ by Eq. 1, then*

$$Q^{(i,\vec{\pi}_{new})}(s, b^{(i)}, \phi^{(i)}) \geq Q^{(i,\vec{\pi})}([s, b^{(i)}], \phi^{(i)})$$

*Proof.* See Appendix C.2. □

The policy evaluation step using Definition 5.1 and policy improvement with Eq. 1, also highlights an important property of agent-chaining: the policy $\pi^{(i)}(\phi^{(i)} \mid s, b^{(i)})$ and Q-values $Q^{(i)}([s, b^{(i)}], \phi^{(i)})$ are both defined on the same domain $[s, b^{(i)}]$ and individual actions $\phi^{(i)}$. While this fact is taken for granted in single-agent RL, this is unlike previous methods in MARL, where using a centralized value function results in a mismatch: e.g. $Q(s, \vec{a})$ is used to update $\pi^{(i)}(a^{(i)} \mid s)$ for simultaneous policy update methods (Lowe et al., 2017; Yu et al., 2022).

We now provide our main theoretical result, that ACPI converges to the optimal policy in both the AC-BMDP as well as the underlying MMDP.

**Theorem 5.4.** *(Agent-Chained Policy Iteration) Starting from any deterministic policy $\vec{\pi} \in \Pi$, the sequence of value functions $\vec{Q}^{\vec{\pi}_n}$ and the improved policies $\vec{\pi}_{n+1}$ converges to the optimal value functions and the policy of the AC-BMDP.*

*i.e. $Q^{(i,*)}([s, b^{(i)}], \phi^{(i)}) = \lim_{n \to \infty} Q^{(i,\vec{\pi}_n)}([s, b^{(i)}], \phi^{(i)}) \geq Q^{(i,\vec{\pi})}([s, b^{(i)}], \phi^{(i)})$ for any $\vec{\pi}, i, s, b^{(i)}, \phi^{(i)}$. Furthermore the optimal policy of the AC-BMDP is also optimal in the underlying MMDP.*

*Proof.* See Appendix C.4. □

The proof for Theorem 5.4 makes use of the fact that there is no loss of generality when considering the space of deterministic action distributions (Corollary C.2). When $\phi^{(i)}$ is deterministic for all agents, the belief $b^{(i)}$ is also deterministic, and the AC-BMDP will reduce to a serialized version of the MMDP.

The full pseudocode for policy iteration is provided in Algorithm 1 in Appendix E. To the best of our knowledge, ACPI is the first policy iteration procedure which converges to the globally optimal policy and naturally extends to practical algorithms in the CTDE setting.

Finally, we note one subtle but important difference when comparing ACPI to MA-PI (Algorithm 1 in Zhong et al. (2024)), which serves as the basis for practical Iterative BR approaches, such as HAPPO and HATRPO. During policy improvement, both MA-PI and ACPI enumerates over all agents $i \in \{1, \ldots N\}$. However, for MA-PI, the policy needs to be improved in sequence, where $\pi^{(i)}$ can only be updated after $\vec{\pi}^{<i}$ has been updated. On the other hand, the result of policy evaluation for ACPI is a set of Q-values $\vec{Q} = \langle Q^{(1,\vec{\pi})}, \ldots Q^{(N,\vec{\pi})} \rangle$ for each agent, and policy improvement for each agent $\pi^{(i)}$ only requires its own $Q^{(i,\vec{\pi})}$. Thus, all agents under ACPI can update their policies in parallel, This difference will prove to be crucial for developing a practical algorithm (ACPO) in the following section.

## 6 AGENT-CHAINED POLICY OPTIMIZATION

Building on the ACPI derived from an AC-BMDP, we introduce Agent-Chained Policy Optimization (ACPO), a practical algorithm designed to approximate the optimal policy which maximizes return.

There are several ways to approximate the policy iteration procedure and derive a practical algorithm applicable to high-dimensional domains. One such method is to use Proximal Policy Optimization (PPO) (Schulman et al., 2017) combined with Generalized Advantage Estimation (GAE) (Schulman et al., 2016), which is a popular choice in single-agent domains.

Following the Bellman operators in Definition 5.1, the Temporal Difference (TD) residual can be written as follows:

$$\zeta_t^{(i)} = \gamma' V^{(i+1)}\left([s_t, b_t^{(i+1)}]\right) - V^{(i)}\left([s_t, b_t^{(i)}]\right), \forall i = \{1, \ldots N-1\}$$

$$\zeta_t^{(N)} = R\left([s_t, b_t^{(N)}], \phi_t^{(N)}\right) + \gamma' V^{(1)}(s_{t+1}) - V^{(N)}\left([s_t, b_t^{(N)}]\right)$$

$$\approx r_t + \gamma' V^{(1)}(s_{t+1}) - V^{(N)}\left([s_t, b_t^{(N)}]\right)$$

where again we have used the fact that the reward $R$ is 0 for any agent $i \in \{1, \ldots N-1\}$.

The advantage is defined as the exponentially weighted sum over the TD residuals,

$$A_t^{(i)} = \sum_{j=i}^{N} (\gamma' \lambda')^{j-i} \zeta_t^{(j)} + \sum_{k=1}^{\infty} \sum_{j=1}^{N} (\gamma' \lambda')^{kN+j-i} \zeta_{t+k}^{(j)},$$

where the detailed derivation is provided in Appendix D.

Using the advantage estimates, the PPO objective can be written as a variant of policy gradient with a clipped probability ratio:

$$\mathcal{L}^{(i)}(\theta) = \mathbb{E}_{\phi_t^{(i)} \sim \pi_{\theta_{old}}^{(i)}(\cdot|s_t, b_t^{(i)})} [\min(w^{(i)}(s_t, b_t^{(i)}, \phi_t^{(i)}) A_t^{(i)}, \ \text{clip}\left(w^{(i)}(s_t, b_t^{(i)}, \phi_t^{(i)}), 1 \pm \epsilon\right) A_t^{(i)})] \tag{2}$$

where $w^{(i)}(s_t, b_t^{(i)}, \phi_t^{(i)}) := \pi_\theta^{(i)}(\phi_t^{(i)}|s_t, b_t^{(i)}) / \pi_{\theta_{old}}^{(i)}(\phi_t^{(i)}|s_t, b_t^{(i)})$.

The objective in Eq. 2 can be further simplified and defined for a policy that produces actions $a^{(i)}$ rather than action distribution $\phi^{(i)}$.

$$\mathcal{L}^{(i)}(\theta) = \mathbb{E}_{a_t^{(i)} \sim \pi_{\theta_{old}}^{(i)}(\cdot|s_t, b_t^{(i)})} [\min(w^{(i)}(s_t, b_t^{(i)}, a_t^{(i)}) A_t^{(i)}, \ \text{clip}\left(w^{(i)}(s_t, b_t^{(i)}, a_t^{(i)}), 1 \pm \epsilon\right) A_t^{(i)})] \tag{3}$$

where $w^{(i)}(s_t, b_t^{(i)}, a_t^{(i)}) := \pi_\theta^{(i)}(a_t^{(i)}|s_t, b_t^{(i)}) / \pi_{\theta_{old}}^{(i)}(a_t^{(i)}|s_t, b_t^{(i)})$. We provide details on the equivalence between Eq. 2 and Eq. 3 in Appendix F.

Overall, we now have a principled PPO-objective derived from an AC-BMDP, with the probability ratio $w^{(i)}$ defined for each individual policy $\pi^{(i)}$. This is unlike previous work, which contains a probability ratio that is the product of $N$ policies (HAPPO) or ignores the product, resulting in a biased objective (MAPPO). For instance, HAPPO and HATRPO contains an importance sampling ratio defined in the form, $w_{\text{BR}}^{(i)}(s, \vec{a}) := \prod_{j=1}^{N} \pi_\theta^{(j)}(a^{(j)}|s) / \prod_{j=1}^{N} \pi_{\theta_{old}}^{(j)}(a^{(j)}|s)$. However, $w_{\text{BR}}^{(i)}$ is problematic as the importance sampling ratio of a product of policies has variance which scales exponentially with the number of agents (Wang et al., 2021b). For MAPPO, the correct importance sampling ratio is also $w_{\text{BR}}^{(i)}$. However, MAPPO simply ignores the product, and thus results in a biased PPO-objective.

It is also worth noting that with PPO as the particular choice, ACPO results in a final objective similar to MAPPO with a few important modifications. Our advantage computation, which uses agent-chaining, is the most crucial change. In the next section, we show that this modification leads to a substantial increase in empirical performance compared to MAPPO, especially in complex domains with many agents.

## 7 EXPERIMENTAL RESULTS

**Environments** We focus our empirical evaluation on Multi-Robot Warehouse (RWARE) (Papoudakis et al., 2021) which simulates a real-world warehouse environment consisting of multiple

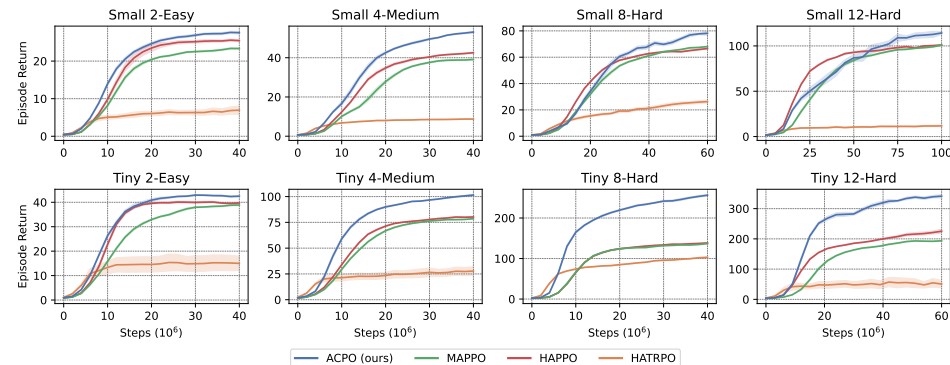

Figure 2: Return for Multi-Robot Warehouse (RWARE) where return is the number of items collected and delivered successfully. The mean and standard error over 10 seeds are reported for all tasks and algorithms except HATRPO and HAPPO on Small 12-Hard (5 seeds) and Tiny 12-Hard (8 seeds).

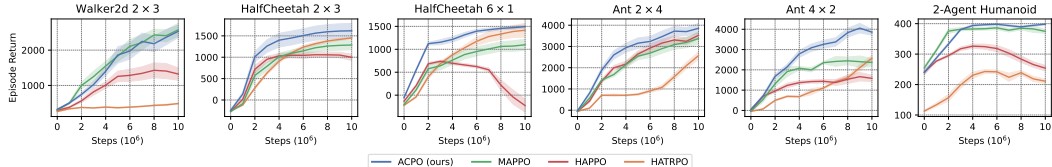

Figure 3: Mean Return and Standard Error over 10 seeds for MA-MuJoCo (Gymnasium).

robots picking up requested shelves and returning them to a designated location. The main challenge in RWARE is *coordination* where the agents must avoid collisions and maximize the number of shelves successfully delivered. We also evaluate our approach on StarCraft Multi-Agent Challenge v2 (SMACv2) (Ellis et al., 2023) and Multi-Agent MuJoCo (Peng et al., 2021), which are popular benchmarks in cooperative MARL with discrete and continuous action spaces, respectively. For SMACv2 and MA-MuJoCo, we closely follow the experimental setup in Zhong et al. (2024).

**Baselines** Our main baselines for ACPO are MAPPO which represents Simultaneous policy update methods and HAPPO/HATRPO which represent Iterative BR methods.[1] MAPPO and HAPPO are the current state-of-the-art on-policy methods in the three domains we consider. Following the baselines considered in Zhong et al. (2024), we also compare against QMIX (Rashid et al., 2018) an off-policy value-based method for discrete action spaces which shows strong performance in SMACv2. QMIX is a value decomposition method which aims to learn individual utility functions which can be aggregated to represent the underlying Q-values.

For a fair comparison, we use the code [2] for all baselines provided in MARLLib (Hu et al., 2023), with the same PPO backbone. For all baselines, we use the reported hyperparameters from Papoudakis et al. (2021) for RWARE, Ellis et al. (2023) for SMACv2 and Zhong et al. (2024) for MA-MuJoCo, and only tune appropriate values when it failed to reproduce the reported performance. For MAPPO, we found the reported hyperparameters to be sufficient for reproducing the results. For ACPO, the same hyperparameters as MAPPO are used for all experiments in order to isolate the effect of agent-chaining, and do not conduct any additional hyperparameter tuning specific to ACPO. We provide details on the hyperparameters in Appendix I.

**Comparative Evaluation** Our main results in Figure 2 show that ACPO outperforms all baselines[3] on all tasks in RWARE, despite having the same backbone PPO and the same hyperparameters

---

[1] We do not compare HATRPO on SMACv2 as it exceeds our computational budget for this work. Moreover, the results from Zhong et al. (2024) showed HATRPO had weaker performance in SMACv2 in comparison to HAPPO, MAPPO and QMIX. Details are provided in Appendix J.

[2] Our anonymous code is available at `https://anonymous.4open.science/r/anonymous-acpo-BD51`.

[3] We also compared with QMIX (Rashid et al., 2018) on RWARE. However, as QMIX failed to learn any meaningful behavior, we do not report their full results. This is consistent with the failure of QMIX on RWARE reported in Papoudakis et al. (2021).

Table 2: Mean return and standard error over 10 seeds on SMACv2.

|  | VDN | QMIX | HAPPO | MAPPO | ACPO (Ours) |
|---|---|---|---|---|---|
| protoss_5_vs_5 | $16.20 \pm 0.49$ | $16.53 \pm 0.55$ | $15.93 \pm 0.54$ | $\mathbf{17.03 \pm 0.92}$ | $\mathbf{18.21 \pm 0.46}$ |
| zerg_5_vs_5 | $11.77 \pm 0.41$ | $\mathbf{14.33 \pm 0.63}$ | $11.81 \pm 0.63$ | $11.84 \pm 0.80$ | $\mathbf{15.16 \pm 0.98}$ |
| protoss_10_vs_11 | $\mathbf{14.74 \pm 0.50}$ | $14.53 \pm 1.06$ | $13.39 \pm 0.50$ | $\mathbf{14.57 \pm 0.33}$ | $\mathbf{15.06 \pm 0.63}$ |
| terran_10_vs_11 | $12.19 \pm 0.60$ | $\mathbf{13.50 \pm 0.73}$ | $10.57 \pm 0.58$ | $12.03 \pm 0.50$ | $\mathbf{13.35 \pm 0.69}$ |
| zerg_10_vs_11 | $13.38 \pm 0.63$ | $\mathbf{14.61 \pm 0.66}$ | $10.77 \pm 0.35$ | $12.48 \pm 0.52$ | $13.20 \pm 0.33$ |

as MAPPO. We also see that the gap widens substantially as the number of agents increases, where the widest gap is seen in 8-agent and 12-agent domains. This provides evidence that ACPO performs substantially better when the environment requires higher levels of coordination. Intuitively, the 12-agent maps require the most coordination among agents since it is the scenario with the most agents crowded in a tight space. Thus, the performance gap jumps even further for the tiny map.

In the results for MA-MuJoCo (Gymnasium) in Figure 3 and SMACv2 in Table 2, ACPO is on par with or outperforms all baselines on all tasks. Notably, ACPO outperforms MAPPO on all tasks with the same hyperparameters, which demonstrates the benefit of agent-chaining. For SMACv2, ACPO outperforms all on-policy baselines, MAPPO and HAPPO. ACPO is also the only on-policy algorithm competitive with QMIX.

**Ablation Results** We ablate the core component of ACPO, which is the advantage computation based on agent chaining. As shown in Figure 4, the variant ACPO without agent chaining can be interpreted as MAPPO augmented with belief states as additional policy inputs. The performance of this variant remains close to MAPPO, indicating that the observed gains of ACPO are not attributable to the extra input, but rather to the agent-chained advantage computation itself.

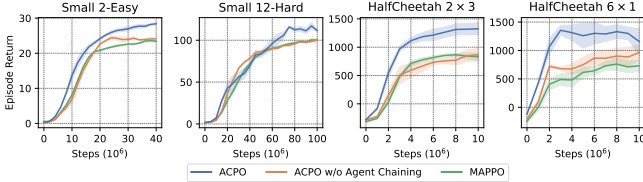
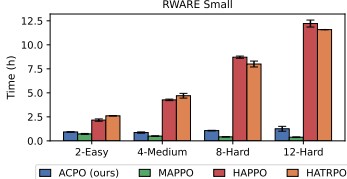

Figure 4: Ablation

Figure 5: Runtime

**Runtime Statistics** In Figure 5, we show the wall-clock training time of running MAPPO, HAPPO, HATRPO and ACPO on RWARE for 5M timesteps. With the number of agents increasing from 2 to 12, the runtime of ACPO remains comparable to MAPPO, and is substantially faster than Iterative BR methods such as HAPPO and HATRPO, which require iteratively updating one agent at a time. Overall, ACPO significantly outperforms baselines in terms of return with only minimal additional computational overhead compared to MAPPO.

## 8 CONCLUSION

In this work, we introduced ACPI, a policy iteration procedure which returns the globally optimal policy for the underlying Multi-Agent MDP (MMDP), as well as ACPO, its practical approximation for Deep MARL. ACPO uses a novel architecture with decentralized critics chained together during centralized training. To the best of our knowledge it is the first algorithm to address the long-standing challenge of going beyond NEs and directly targeting convergence to the globally optimal policy under CTDE. Theoretical insights are reflected in experimental results, where we showed substantial increase in performance, especially in complex domains with many agents.

In single-agent RL, the development of principled yet practical algorithms such as PPO and SAC have led to the wide application of RL in areas such as self-driving cars (Kiran et al., 2022) and Large Language Models (LLMs) (Ouyang et al., 2022), among many others. Similarly, we hope that ACPO forms the basis for better algorithms as well as a wider adoption of MARL to various domains, including Multi-Agent LLMs (Wu et al., 2024; Liu et al., 2025).

## THE USE OF LARGE LANGUAGE MODELS (LLMS)

In this project, LLMs were used solely as an assist tool for improving the readability of this manuscript. All ideas, proofs, and analyses are by the authors. The authors have verified and edited all content generated by LLMs.

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

## A  BELIEF UPDATE

$$\tau\left([s_t, b_t^{(i)}], \phi_t^{(i)}\right)(\vec{a}_t^{<i+1}) = \frac{1}{\eta([s_t, b_t^{(i)}], \phi_t^{(i)})} \sum_{\vec{a}_t^{<i}} b_t^{(i)}(\vec{a}_t^{<i}) T\left([s_t, \vec{a}_t^{<i+1}] | [s_t, \vec{a}_t^{<i}], \phi_t^{(i)}\right)$$

$$= \frac{1}{\eta([s_t, b_t^{(i)}], \phi_t^{(i)})} \sum_{\vec{a}_t^{<i}} b_t^{(i)}(\vec{a}_t^{<i}) \sum_{a_t^{(i)}} \phi_t^{(i)}(a_t^{(i)}) T\left([s_t, \vec{a}_t^{<i+1}] | [s_t, \vec{a}_t^{<i}], a_t^{(i)}\right)$$

Finally, $\eta$ is the normalization factor defined as

$$\eta([s_t, b_t^{(i)}], \phi_t^{(i)}) = \sum_{\vec{a}_t^{<i+1}} \sum_{\vec{a}_t^{<i}} b_t^{(i)}(\vec{a}_t^{<i}) \sum_{a_t^{(i)}} \phi_t^{(i)}(a_t^{(i)}) T\left([s_t, \vec{a}_t^{<i+1}] | [s_t, \vec{a}_t^{<i}], a_t^{(i)}\right)$$

## B  PROOF FOR THEOREM 3.1

The serialized multi-agent problem can be described as follows:

- Action space $A^{(i)}$ of individual actions for some $i \in \mathcal{N}$
- States $[s_t, \vec{a}_t^{<i}]$ which augments the original state space with actions selected by previous agents $\vec{a}_t^{<i}$.
- Reward function
$$R([s_t, \vec{a}_t^{<i}], a_t^{(i)}) = \begin{cases} R(s_t, \vec{a}_t) & \text{if } i = N \\ 0 & \text{otherwise} \end{cases}$$

- Transition function
$$T([s_{t+1}, \emptyset] \mid [s_t, \vec{a}_t^{<i}], a_t^{(i)}) = T(s_{t+1} \mid s_t, \vec{a}_t) \text{ if } i = N$$
$$T([s_t, \vec{a}_t^{<i+1}] \mid [s_t, \vec{a}_t^{<i}], a_t^{(i)})$$
$$= \mathbb{I}\{[\vec{a}_t^{<i}, a_t^{(i)}] = \vec{a}_t^{<i+1}\} \text{ if } i \in \{1, \ldots, N-1\}$$

- Discount factor $\gamma'$ where $\gamma' = \gamma^{1/N}$

**Theorem 3.1.** *(Peralez et al., 2025) For every MMDP, there exists a serialized multi-agent problem, of which its optimal policy is also optimal for the underlying MMDP.*

*Proof.* Let $\vec{\pi}^* = \langle \pi^{(1,*)}, \ldots, \pi^{(N,*)} \rangle$ be the optimal policy over the serialized MMDP.

$$V^{(i,*)}([s, \vec{a}^{<i}])$$

$$= \mathbb{E}\left[\sum_{j=i}^{N} (\gamma')^{j-i} R\left(s_0, \vec{a}_0^{<j}, a_0^{(j)}\right) \mid s_0 = s, \vec{a}_0^{<i} = \vec{a}^{<i}, \vec{\pi}^*\right] + \mathbb{E}\left[\sum_{t=1}^{N} \sum_{j=1}^{N} (\gamma')^{tN+j-i} R\left(s_t, \vec{a}_t^{<j}, a_t^{(j)}\right) \mid \vec{\pi}^*\right]$$

$$= \mathbb{E}\left[(\gamma')^{N-i} R\left(s_0, \vec{a}_0^{<N}, a_0^{(N)}\right) \mid s_0 = s, \vec{a}_0^{<i} = \vec{a}^{<i}, \vec{\pi}^*\right] + \mathbb{E}\left[\sum_{t=1} (\gamma')^{tN+N-i} R\left(s_t, \vec{a}_t^{<N}, a_t^{(N)}\right) \mid \vec{\pi}^*\right]$$

$$= \mathbb{E}\left[(\gamma')^{N-i} R\left(s_0, \vec{a}_0^{<N}, a_0^{(N)}\right) \mid s_0 = s, \vec{a}_0^{<i} = \vec{a}^{<i}, \vec{\pi}^*\right] + \mathbb{E}\left[\sum_{t=1} (\gamma')^{N-i} \gamma^t R\left(s_t, \vec{a}_t^{<N}, a_t^{(N)}\right) \mid \vec{\pi}^*\right]$$

$$= \mathbb{E}\left[\sum_{t=0} (\gamma')^{N-i} \gamma^t R\left(s_t, \vec{a}_t^{<N}, a_t^{(N)}\right) \mid s_0 = s, \vec{a}_0^{<i} = \vec{a}^{<i}, \vec{\pi}^*\right]$$

$$= (\gamma')^{N-i} \mathbb{E}\left[\sum_{t=0} \gamma^t R\left(s_t, \vec{a}_t^{<N}, a_t^{(N)}\right) \mid s_0 = s, \vec{a}_0^{<i} = \vec{a}^{<i}, \vec{\pi}^*\right]$$

$$\tag{4}$$

$$V^{(1,*)}(s) = (\gamma')^{N-1} \mathbb{E}\left[\sum_{t=0} \gamma^t R\left(s_t, \vec{a}_t^{<N}, a_t^{(N)}\right) \mid \vec{\pi}^*\right] \text{ for agent 1.}$$

$$V^{(N,*)}([s, \vec{a}^{<N}]) = \mathbb{E}\left[\sum_{t=0} \gamma^t R\left(s_t, \vec{a}_t^{<N}, a_t^{(N)}\right) \mid s_0 = s, \vec{a}_0^{<N} = \vec{a}^{<N}, \vec{\pi}^*\right] \text{ for agent } N.$$

Thus, we have established that the optimal policy $\vec{\pi}^*$ in the serialized problem will obtain the same expected value in the MMDP (times a constant factor). The same holds for any policy $\vec{\pi}$, and thus there is a 1-1 mapping between serialized and simultaneous policies which yield the same value.

$\square$

## C  PROOFS FOR POLICY ITERATION CONVERGENCE

### C.1  PROOF FOR LEMMA 5.2

**Lemma 5.2.** *(Agent-Chained Policy Evaluation) The Agent-Chained Bellman Operators in Definition 5.1 are a contraction mapping under the infinity norm. Thus, starting with any $\vec{Q} = \langle Q^{(1)}, \ldots Q^{(N)} \rangle$ and a joint policy $\vec{\pi} = \langle \pi^{(1)}, \ldots, \pi^{(N)} \rangle$, the repeated application of $\mathcal{T}^{\vec{\pi}}$ will return a set of Q-values for each agent $\langle Q^{(1,\vec{\pi})}, \ldots Q^{(N,\vec{\pi})} \rangle$ in the limit.*

*Proof.* First, note that we can view $\langle Q^{(1)}, \ldots, Q^{(N)} \rangle$ as a single Q-function with the state space further augmented by agent ID. Under this perspective, we now have a single policy denoted as $\pi$ and a corresponding value function $Q^\pi$, defined on the AC-BMDP.

Since the AC-BMDP is a single-agent Belief MDP, the rest follows standard convergence results of policy evaluation (Agarwal et al., 2019), which we include for completeness.

For any agent $i \in \{1, \ldots N-1\}$, state $[s, b^{(i)}, i]$, action $\phi^{(i)}$ and arbitrary Q-values $Q_1, Q_2$,

$$\left| \mathcal{T}^\pi Q_1([s, b^{(i)}, i], \phi^{(i)}) - \mathcal{T}^\pi Q_2([s, b^{(i)}, i], \phi^{(i)}) \right|$$

$$= \left| \mathbb{E}_{\substack{[s, b^{(i+1)}, i+1] = T([s, b^{(i)}, i], \phi^{(i)}) \\ \phi^{(i+1)} \sim \pi(\cdot | s, b^{(i+1)}, i+1)}} \left[ \gamma' Q_1([s, b^{(i+1)}, i+1], \phi^{(i+1)}) - \gamma' Q_2([s, b^{(i+1)}, i+1], \phi^{(i+1)}) \right] \right|$$

$$\leq \gamma' \max_{\phi^{(i+1)}} \left| Q_1([s, b^{(i+1)}, i+1], \phi^{(i+1)}) - Q_2([s, b^{(i+1)}, i+1], \phi^{(i+1)}) \right|$$

$$\leq \gamma' \max_{\phi^{(i+1)}, b^{(i+1)}, j \in \{1, \ldots, N\}} \left| Q_1([s, b^{(i+1)}, j], \phi^{(i+1)}) - Q_2([s, b^{(i+1)}, j], \phi^{(i+1)}) \right|$$

For agent $N$,

$$\left| \mathcal{T}^\pi Q_1([s, b^{(N)}, N], \phi^{(N)}) - \mathcal{T}^\pi Q_2([s, b^{(N)}, N], \phi^{(N)}) \right|$$

$$= \left| \mathbb{E}_{\substack{[s', 1] \sim T(\cdot | [s, b^{(N)}, N], \phi^{(N)}) \\ \phi^{(1)} \sim \pi(\cdot | s', 1)}} \left[ \gamma' Q_1([s', 1], \phi^{(1)}) - \gamma' Q_2([s', 1], \phi^{(1)}) \right] \right|$$

$$\leq \gamma' \max_{s', \phi^{(1)}} \left| Q_1([s', 1], \phi^{(1)}) - Q_2([s', 1], \phi^{(1)}) \right|$$

$$\leq \gamma' \max_{s', \phi^{(1)}, j \in \{1, \ldots, N\}} \left| Q_1([s', j], \phi^{(1)}) - Q_2([s', j], \phi^{(1)}) \right|$$

Thus, $\mathcal{T}^\pi$ is a contraction mapping under the infinity norm, i.e. there exists $\gamma' \in [0, 1)$ such that

$$\|\mathcal{T}^\pi Q_1 - \mathcal{T}^\pi Q_2\|_\infty \leq \gamma' \|Q_1 - Q_2\|_\infty$$

Since $\mathcal{T}^\pi$ is a contraction mapping, we have the following:
$$\|Q_k - Q^\pi\|_\infty = \|\mathcal{T}^\pi Q_{k-1} - \mathcal{T}^\pi Q^\pi\|_\infty$$
$$\leq \gamma' \|Q_{k-1} - Q^\pi\|_\infty$$
$$\vdots$$
$$\leq (\gamma')^k \|Q_0 - Q^\pi\|_\infty$$

If we let $k \to \infty$, $\|Q_k - Q^\pi\|_\infty = 0$, and $\lim_{k\to\infty} Q_k = Q^\pi$. By the Banach fixed-point theorem, this solution is unique.

$\square$

## C.2 PROOF FOR LEMMA 5.3

**Lemma 5.3.** *(Agent-Chained Policy Improvement) Given a deterministic policy $\vec{\pi} = \langle \pi^{(1)}, \ldots, \pi^{(N)} \rangle$, let $Q^{(i,\vec{\pi})}$ denote the $i$-th agent's value function for a joint policy $\vec{\pi}$. If we update the new policy $\vec{\pi}_{new} = \langle \pi_{new}^{(1)}, \ldots, \pi_{new}^{(N)} \rangle$ by Eq. 1, then*

$$Q^{(i,\vec{\pi}_{new})}(s, b^{(i)}, \phi^{(i)}) \geq Q^{(i,\vec{\pi})}([s, b^{(i)}], \phi^{(i)})$$

*Proof.* As in the proof for Lemma 5.2, we consider $\vec{\pi}$ to be a single policy $\pi$ which is augmented by agent ID in the state space.

For any $i \in \{1, \ldots N - 1\}, s, b^{(i)}, \phi^{(i)}$,

$$Q^\pi([s, b^{(i)}, i], \phi^{(i)}) = \gamma' \mathbb{E}_{\substack{[s,b^{(i+1)},i+1]=T([s,b^{(i)},i],\phi^{(i)}) \\ \phi^{(i+1)} \sim \pi(\cdot|[s,b^{(i+1)},i+1])}} \left[ Q^\pi \left( [s, b^{(i+1)}, i+1], \phi^{(i+1)} \right) \right]$$

$$\leq \gamma' \mathbb{E}_{[s,b^{(i+1)},i+1]=T([s,b^{(i)},i],\phi^{(i)})} \left[ \max_{\phi^{(i+1)}} Q^\pi \left( [s, b^{(i+1)}, i+1], \phi^{(i+1)} \right) \right]$$

$$= \gamma' \mathbb{E}_{\substack{[s,b^{(i+1)},i+1]=T([s,b^{(i)},i],\phi^{(i)}) \\ \phi^{(i+1)} \sim \pi_{new}(\cdot|[s,b^{(i+1)},i+1])}} \left[ Q^\pi \left( [s, b^{(i+1)}, i+1], \phi^{(i+1)} \right) \right]$$

For $i = N$ and any $s, b^{(i)}, \phi^{(i)}$,

$$Q^\pi([s, b^{(N)}, N], \phi^{(N)}) = R([s, b^{(N)}, N], \phi^{(N)}) + \gamma' \mathbb{E}_{\substack{[s',1]=T([s,b^{(N)},N],\phi^{(N)}) \\ \phi^{(1)} \sim \pi(\cdot|[s',1])}} \left[ Q^\pi \left( [s', 1], \phi^{(1)} \right) \right]$$

$$\leq R([s, b^{(N)}, N], \phi^{(N)}) + \gamma' \mathbb{E}_{[s',1]=T([s,b^{(N)},N],\phi^{(N)}))} \left[ \max_{\phi^{(1)}} Q^\pi \left( [s', 1], \phi^{(1)} \right) \right]$$

$$= R([s, b^{(N)}, N], \phi^{(N)}) + \gamma' \mathbb{E}_{\substack{[s',1]=T([s,b^{(N)},N],\phi^{(N)}) \\ \phi^{(1)} \sim \pi_{new}(\cdot|[s',1])}} \left[ Q^\pi \left( [s', 1], \phi^{(1)} \right) \right]$$

Thus, for any $i \in \{1, \ldots N\}, s, b^{(i)}, \phi^{(i)}$,

$$Q^\pi([s, b^{(i)}, i], \phi^{(i)}) \leq R([s, b^{(i)}, i], \phi^{(i)}) + \gamma' \mathbb{E}_{\substack{[s,b^{(i+1)},i+1]=T([s,b^{(i)},i],\phi^{(i)}) \\ \phi^{(i+1)} \sim \pi_{new}(\cdot|[s,b^{(i+1)},i+1])}} \left[ Q^\pi \left( [s, b^{(i+1)}, i+1], \phi^{(i+1)} \right) \right]$$

$$\vdots$$

$$\leq Q^{\pi_{new}}([s, b^{(i)}, i], \phi^{(i)})$$

$\square$

## C.3 CHARACTERIZATION OF $Q^{(i,*)}$

Before our main result, we provide a useful property of the optimal Q-values of the AC-BMDP, which has an affine structure.

**Theorem C.1.** *Let $Q^{(i,*)}$ denote the $i$-th agent's Q-values for the optimal policy under an AC-BMDP. Then, $Q^{(i,*)}$ is an affine function of $\phi^{(i)}$. i.e. for all $i \in \{1, \ldots, N\}, s, b^{(i)}, \phi^{(i)}$,*

$$Q^{(i,*)}([s, b^{(i)}], \phi^{(i)}) = \sum_{a^{(i)}} \phi^{(i)}(a^{(i)}) Q^{(i,*)}([s, b^{(i)}], \delta_{a^{(i)}})$$

*where $\delta_{a^{(i)}}$ denotes a particular action distribution which deterministically selects $a^{(i)}$.*

*Proof.* We prove the claim by induction.

For any $s, b^{(N)}, \phi^{(N)}$ at terminal timesteps,

$$Q^{(N)}([s, b^{(N)}], \delta_{a^{(N)}}) = R([s, b^{(N)}], a^{(N)})$$

$$\begin{aligned} Q^{(N)}([s, b^{(N)}], \phi^{(N)}) &= R([s, b^{(N)}], \phi^{(N)}) \\ &= \sum_{a^{(N)}} \phi^{(N)}(a^{(N)}) R([s, b^{(N)}], a^{(N)}) \\ &= \sum_{a^{(N)}} \phi^{(N)}(a^{(N)}) Q^{(N)}([s, b^{(N)}], \delta_{a^{(N)}}) \end{aligned}$$

$$Q^{(N)}([s, b^{(N)} = \vec{a}^{<N}], \phi^{(N)}) = R([s, \vec{a}^{<N}], \phi^{(N)})$$

$$\begin{aligned} Q^{(N)}([s, b^{(N)}], \phi^{(N)}) &= R([s, b^{(N)}], \phi^{(N)}) \\ &= \sum_{\vec{a}^{<N}} b^{(N)}(\vec{a}^{<N}) R([s, \vec{a}^{<N}], \phi^{(N)}) \\ &= \sum_{\vec{a}^{<N}} b^{(N)}(\vec{a}^{<N}) Q^{(N)}([s, \vec{a}^{<N}], \phi^{(N)}) \end{aligned}$$

where $\vec{a}^{<N}$ in $Q^{(N)}([s, \vec{a}^{<N}], \phi^{(N)})$ denotes the belief that assigns probability 1 to $\vec{a}^{<N}$.

For agents $i \in \{1, \ldots, N-1\}$,

$$\begin{aligned} Q^*([s, b^{(i)}], \delta_{a^{(i)}}) &= \gamma' \max_{\phi^{(i+1)}} Q^*([s, b^{(i+1)}], \phi^{(i+1)}) \\ &= \gamma' \max_{\phi^{(i+1)}} \sum_{\vec{a}^{<i+1}} b^{(i+1)}(\vec{a}^{<i+1}) Q^*([s, \vec{a}^{<i+1}], \phi^{(i+1)}) \text{(by induction)} \\ &= \gamma' \max_{\phi^{(i+1)}} \frac{1}{\eta([s, b^{(i)}], \phi^{(i)})} \sum_{\vec{a}^{<i}} b^{(i)}(\vec{a}^{<i}) Q^*([s, \vec{a}^{<i}, a^{(i)}], \phi^{(i+1)}) \end{aligned}$$

where $[s, b^{(i+1)}] = T([s, b^{(i)}], \delta_{a^{(i)}})$.

$$\begin{aligned} Q^*([s, b^{(i)}], \phi^{(i)}) &= \gamma' \max_{\phi^{(i+1)}} Q^*([s, b^{(i+1)}], \phi^{(i+1)}) \\ &= \gamma' \max_{\phi^{(i+1)}} \sum_{\vec{a}^{<i+1}} b^{(i+1)}(\vec{a}^{<i+1}) Q^*([s, \vec{a}^{<i+1}], \phi^{(i+1)}) \\ &= \gamma' \max_{\phi^{(i+1)}} \frac{1}{\eta([s, b^{(i)}], \phi^{(i)})} \sum_{\vec{a}^{<i}} b^{(i)}(\vec{a}^{<i}) \sum_{a^{(i)}} \phi^{(i)}(a^{(i)}) Q^*([s, \vec{a}^{<i}, a^{(i)}], \phi^{(i+1)}) \\ &= \gamma' \max_{\phi^{(i+1)}} \frac{1}{\eta([s, b^{(i)}], \phi^{(i)})} \sum_{\vec{a}^{<i}} b^{(i)}(\vec{a}^{<i}) Q^*([s, \vec{a}^{<i}, a^{(i)}], \phi^{(i+1)}) \\ &= \sum_{a^{(i)}} \phi^{(i)}(a^{(i)}) Q^*([s, b^{(i)}], \delta_{a^{(i)}}) \end{aligned}$$

where $[s, b^{(i+1)}] = T([s, b^{(i)}], \phi^{(i)})$. $\square$

A direct corollary of Theorem C.1 is that only the actions $a^{(i)}$ need to be enumerated rather than the full space of 1-step policies $\phi^{(i)}$.

**Corollary C.2.** *For all $i \in \{1, \ldots, N\}, s, b^{(i)}$,*

$$\max_{\phi^{(i)}} Q^{(i,*)} \left( [s, b^{(i)}], \phi^{(i)} \right) = \max_{a^{(i)}} Q^{(i,*)} \left( [s, b^{(i)}], \delta_{a^{(i)}} \right)$$

## C.4   PROOF FOR THEOREM 5.4

**Theorem 5.4.** *(Agent-Chained Policy Iteration) Starting from any deterministic policy* $\vec{\pi} \in \Pi$, *the sequence of value functions* $\vec{Q}^{\vec{\pi}_n}$ *and the improved policies* $\vec{\pi}_{n+1}$ *converges to the optimal value functions and the policy of the AC-BMDP.*

*i.e.* $Q^{(i,*)}([s, b^{(i)}], \phi^{(i)}) = \lim_{n \to \infty} Q^{(i,\vec{\pi}_n)}([s, b^{(i)}], \phi^{(i)}) \geq Q^{(i,\vec{\pi})}([s, b^{(i)}], \phi^{(i)})$ *for any* $\vec{\pi}, i, s, b^{(i)}, \phi^{(i)}$. *Furthermore the optimal policy of the AC-BMDP is also optimal in the underlying MMDP.*

*Proof.* By the monotonic improvement property in Lemma 5.3, we know that for any $i \in \{1, \ldots, N\}, b^{(i)}, s, \phi^{(i)}$,

$$Q^{\vec{\pi}_{n+1}}([s, b^{(i)}], \phi^{(i)}) \geq Q^{\vec{\pi}_n}([s, b^{(i)}], \phi^{(i)})$$

.

If there is no improvement,

$$
\begin{aligned}
Q^{\vec{\pi}_n}([s, b^{(i)}], \phi^{(i)}) &= Q^{\vec{\pi}_{n+1}}([s, b^{(i)}], \phi^{(i)}) \\
&= \gamma' Q^{\vec{\pi}_{n+1}}([s, b^{(i+1)}], \pi_{n+1}^{(i)}([s, b^{(i+1)}])) \\
&= \gamma' Q^{\vec{\pi}_n}([s, b^{(i+1)}], \pi_{n+1}^{(i)}([s, b^{(i+1)}])) \\
&= \gamma' \max_{\phi^{(i+1)}} Q^{\vec{\pi}_n}([s, b^{(i+1)}], \phi^{(i+1)}))
\end{aligned}
$$

where $[s, b^{(i+1)}] = T([s, b^{(i)}], \phi^{(i)})$. Thus, at the limit $\lim_{n \to \infty} Q^{\vec{\pi}_n}([s, b^{(i)}], \phi^{(i)})$, the Bellman optimality equations are satisfied.

Due to Corollary C.2, it is sufficient to consider the following policy improvement procedure considering only the $\delta_{a^{(i)}}$, which is the space of deterministic $\phi^{(i)}$:

$$\forall i, b^{(i)}, s, \pi_{new}^{(i)}([s, b^{(i)}]) \leftarrow \arg\max_{a^{(i)}} Q^{\vec{\pi}}([s, b^{(i)}], \delta_{a^{(i)}}) \tag{5}$$

Note that if we restrict ourselves to an AC-BMDP defined over the space of deterministic 1-step policies $\delta_{a^{(i)}}$, all of the components in the AC-BMDP are equivalent to that of the serialized version of the MMDP. $\qquad\square$

## D   DETAILS ON ADVANTAGE COMPUTATION

The advantage can be written as an exponentially-weighted sum over the TD residuals,

$$A_t^{(1)} = \sum_{j=1}^{N} (\gamma'\lambda')^{j-1} \zeta_t^{(j)} + \sum_{k=1}^{\infty} \sum_{j=1}^{N} (\gamma'\lambda')^{kN+j-1} \zeta_{t+k}^{(j)}$$

$$A_t^{(2)} = \sum_{j=2}^{N} (\gamma'\lambda')^{j-2} \zeta_t^{(j)} + \sum_{k=1}^{\infty} \sum_{j=1}^{N} (\gamma'\lambda')^{kN+j-2} \zeta_{t+k}^{(j)}$$

$$\vdots$$

$$A_t^{(N)} = \zeta_t^{(N)} + \sum_{k=1}^{\infty} \sum_{j=1}^{N} (\gamma'\lambda')^{kN+j-N} \zeta_{t+k}^{(j)}$$

$$\therefore A_t^{(i)} = \sum_{j=i}^{N} (\gamma'\lambda')^{j-i} \zeta_t^{(j)} + \sum_{k=1}^{\infty} \sum_{j=1}^{N} (\gamma'\lambda')^{kN+j-i} \zeta_{t+k}^{(j)}.$$

# E  PSEUDOCODES

We first present the pseudocode for ACPI which is defined directly on the action space consisting of 1-step policies. $\phi^{(i)}$. This is a straightforward policy iteration procedure defined on the AC-BMDP.

---

**Algorithm 1** Agent-Chained Policy Iteration

---

1: Randomly initialize $\vec{\pi} = \left(\pi^{(1)}, \ldots, \pi^{(N)}\right)$ and $\vec{Q} = \left(Q^{(1)}, \ldots, Q^{(N)}\right)$.
2: **while** $\vec{\pi}$ not converged **do**
3:    **while** $\vec{Q}$ not converged **do**
4:      # Policy Evaluation
5:

$$\forall i \in \{1, \ldots, N-1\}, s, b^{(i)}, \phi^{(i)},$$

$$Q^{(i)}([s, b^{(i)}], \phi^{(i)}) \leftarrow \gamma' \mathbb{E}_{\substack{b^{(i+1)} = T([s, b^{(i)}], \phi^{(i)}) \\ \phi^{(i+1)} \sim \pi^{(i+1)}(\cdot | s, b^{(i+1)})}} \left[ Q^{(i+1)}\left([s, b^{(i+1)}], \phi^{(i+1)}\right) \right]$$

6:

$$\forall s, b^{(N)}, \phi^{(N)},$$

$$Q^{(N)}([s, b^{(N)}], \phi^{(N)})$$

$$\leftarrow R\left([s, b^{(N)}], \phi^{(N)}\right) + \gamma' \mathbb{E}_{\substack{s' \sim T(\cdot | [s, b^{(N)}], \phi^{(N)}) \\ \phi^{(1)} \sim \pi^{(1)}(\cdot | s')}} \left[ Q^{(1)}\left(s', \phi^{(1)}\right) \right]$$

7:    **end while**
8:    # Policy Improvement
9:

$$\forall i \in \{1, \ldots, N\}, s, b^{(i)},$$

$$\pi^{(i)}(s, b^{(i)}) \leftarrow \arg\max_{\phi^{(i)}} Q^{(i)}\left([s, b^{(i)}], \phi^{(i)}\right)$$

10: **end while**

---

As we showed in Corollary C.2, the AC-BMDP has a special structure which ensures that it is sufficient to consider the space of deterministic $\phi^{(i)}$ for finding an optimal policy. Thus, we can also define an equivalent policy iteration procedure over the action space $\delta_{a^{(i)}}$ (Algorithm 2).

Finally, we present ACPO which is a practical algorithm that aims to approximate ACPI via the PPO objective.

We reiterate below the definition of the TD residual.

$$\zeta_t^{(i)} = \gamma' V^{(i+1)}\left([s_t, b_t^{(i+1)}]\right) - V^{(i)}\left([s_t, b_t^{(i)}]\right), \forall i = \{1, \ldots N-1\}$$

$$\zeta_t^{(N)} = R\left([s_t, b_t^{(N)}], \phi_t^{(N)}\right) + \gamma' V^{(1)}(s_{t+1}) - V^{(N)}\left([s_t, b_t^{(N)}]\right)$$

$$\approx r_t + \gamma' V^{(1)}(s_{t+1}) - V^{(N)}\left([s_t, b_t^{(N)}]\right)$$

---

**Algorithm 2** Agent-Chained Policy Iteration (Deterministic action distribution $\phi^{(i)}$)

---

1: Randomly initialize $\vec{\pi} = \left(\pi^{(1)}, \ldots, \pi^{(N)}\right)$ and $\vec{Q} = \left(Q^{(1)}, \ldots, Q^{(N)}\right)$.
2: **while** $\vec{\pi}$ not converged **do**
3:    **while** $\vec{Q}$ not converged **do**
4:      # Policy Evaluation
5:

$$\forall i \in \{1, \ldots, N-1\}, s, \vec{a}^{<i}, a^{(i)},$$

$$Q^{(i)}([s, b^{(i)} = \vec{a}^{<i}], \delta_{a^{(i)}}) \leftarrow \gamma' \mathbb{E}_{\substack{b^{(i+1)} = T([s, b^{(i)} = \vec{a}^{<i}], \delta_{a^{(i)}}) \\ \delta_{a^{(i+1)}} \sim \pi^{(i+1)}(\cdot | s, b^{(i+1)})}} \left[ Q^{(i+1)}\left([s, b^{(i+1)}], \delta_{a^{(i+1)}}\right) \right]$$

6:

$$\forall s, \vec{a}^{<N}, a^{(N)},$$

$$Q^{(N)}([s, b^{(N)} = \vec{a}^{<N}], \delta_{a^{(N)}})$$

$$\leftarrow R\left([s, \vec{a}^{<N}], a^{(N)}\right) + \gamma' \mathbb{E}_{\substack{s' \sim T(\cdot | [s, b^{(N)} = \vec{a}^{<N}], \delta_{a^{(N)}}) \\ \delta_{a^{(1)}} \sim \pi^{(1)}(\cdot | s')}} \left[ Q^{(1)}\left(s', \delta_{a^{(1)}}\right) \right]$$

7:    **end while**
8:    # Policy Improvement
9:

$$\forall i \in \{1, \ldots, N\}, s, \vec{a}^{<i},$$

$$\pi^{(i)}(s, \vec{a}^{<i}) \leftarrow \arg\max_{a^{(i)}} Q^{(i)}\left([s, \vec{a}^{<i}], \delta_{a^{(i)}}\right)$$

10: **end while**

---

---

**Algorithm 3** Agent-Chained Policy Optimization (PPO-based)

---

1: **Initialize**: Actor networks $\theta_0 = [\theta_0^{(1)}, \ldots, \theta_0^{(N)}]$, Critic networks $\vec{\psi}_0 = [\psi_0^{(1)}, \ldots, \psi_0^{(N)}]$.
2: **while** $t \leq t_{max}$ **do**
3:    Collect transitions $\left(s_t, \{a_t^{(i)}\}_{i=1}^N, r_t, s_{t+1}\right)$ by running the joint policy $\vec{\pi}_{\theta_k}$.
4:    Compute advantages after each episode: $\forall i = 1, \ldots, N$,

$$A_t^{(i)} = \sum_{j=i}^{N} (\gamma' \lambda')^{j-i} \zeta_t^{(j)} + \sum_{k=1}^{\infty} \sum_{j=1}^{N} (\gamma' \lambda')^{kN+j-i} \zeta_{t+k}^{(j)}$$

5:    Update actors with the PPO-Clip objective: $\forall i = 1 \ldots, N$,

$$\mathbb{E}_{a_t^{(i)} \sim \pi_{\theta_{old}}^{(i)}(\cdot | s_t, b_t^{(i)})} [\min(w^{(i)}(s_t, b_t^{(i)}, a_t^{(i)}) A_t^{(i)}, \; \text{clip}\left(w^{(i)}(s_t, b_t^{(i)}, a_t^{(i)}), 1 - \epsilon, 1 + \epsilon\right) A_t^{(i)})]$$

     where $w^{(i)}(s_t, b_t^{(i)}, a_t^{(i)}) := \frac{\pi_\theta^{(i)}(a_t^{(i)} | s_t, b_t^{(i)})}{\pi_{\theta_{old}}^{(i)}(a_t^{(i)} | s_t, b_t^{(i)})}$.
6:    Update Decentralized Critics: $\forall i = 1 \ldots, N$,

$$\psi_{k+1}^{(i)} = \arg\min_{\psi^{(i)}} \mathbb{E}\left[ \left( V_\phi^{(i)}(s_t) - \hat{R}_t^{(i)} \right)^2 \right]$$

7: **end while**

---

# F   FINAL PPO OBJECTIVE

To simplify notation, we provide the derivation for agent 1 which does not have belief $b^{(i)}$ in its state space, and assume that the parameters $\theta$ are shared across agents.

$$\nabla_\theta J(\theta) = \mathbb{E}_{\phi^{(1)} \sim \pi_\theta^{(1)}(\cdot|s)} \left[ \nabla_\theta \log \pi_\theta^{(1)}(\phi^{(1)}|s) Q^{(1)}(s, \phi^{(1)}) \right]$$

$$= \int_{\phi^{(1)}} \pi_\theta^{(1)}(\phi^{(1)}|s) \nabla_\theta \log \pi_\theta^{(1)}(\phi^{(1)}|s) Q^{(1)}(s, \phi^{(1)}) d\phi^{(1)}$$

$$= \int_{\phi^{(1)}} \pi_\theta^{(1)}(\phi^{(1)}|s) \nabla_\theta \log \pi_\theta^{(1)}(\phi^{(1)}|s) \int_{a^{(1)}} \phi^{(1)}(a^{(1)}) Q^{(1)}(s, a^{(1)}) da^{(1)} d\phi^{(1)}$$

$$= \int_{a^{(1)}} \int_{\phi^{(1)}} \pi_\theta^{(1)}(\phi^{(1)}|s) \nabla_\theta \log \pi_\theta^{(1)}(\phi^{(1)}|s) \phi^{(1)}(a^{(1)}) d\phi^{(1)} \, Q^{(1)}(s, a^{(1)}) da^{(1)}$$

$$= \int_{a^{(1)}} \int_{\phi^{(1)}} \nabla_\theta \pi_\theta^{(1)}(\phi^{(1)}|s) \phi^{(1)}(a) d\phi^{(1)} \, Q^{(1)}(s, a^{(1)}) da^{(1)}$$

$$= \int_{a^{(1)}} \nabla_\theta \left( \underbrace{\int_{\phi^{(1)}} \pi_\theta^{(1)}(\phi^{(1)}|s) \phi^{(1)}(a^{(1)}) d\phi^{(1)}}_{= \pi_\theta^{(1)}(a^{(1)}|s)} \right) Q^{(1)}(s, a^{(1)}) da^{(1)}$$

$$= \int_{a^{(1)}} \nabla_\theta \pi_\theta^{(1)}(a^{(1)}|s) Q(s, a^{(1)}) da^{(1)}$$

$$= \int_{a^{(1)}} \pi_\theta^{(1)}(a^{(1)}|s) \nabla_\theta \log \pi_\theta^{(1)}(a^{(1)}|s) Q(s, a^{(1)}) da^{(1)}$$

$$= \mathbb{E}_{a^{(1)} \sim \pi_\theta^{(1)}(a^{(1)}|s)} \left[ \nabla_\theta \log \pi_\theta^{(1)}(a^{(1)}|s) Q(s, a^{(1)}) \right]$$

We can obtain the same conclusion for other agents through a similar derivation.

## G    EXACT CALCULATION OF POLICIES IN THE MATRIX GAME

|   | A | B | C |
|---|---|---|---|
| A | 5 | $-20$ | $-20$ |
| B | $-20$ | 10 | $-20$ |
| C | $-20$ | $-20$ | 20 |

Figure 6: 3x3 Matrix Game

Here we provide details on how ACPO can solve the Matrix Game provided in Table 1 and repeated above in Figure 6.

Due to serialization, ACPO considers this as a 2-step game even though the underlying game is a 1-step game. Since we are in a simple toy setting which can be solved by policy iteration, we only consider deterministic $\phi$.

Policy evaluation for agent 1 is conducted as follows.

$$Q^{(1)}(\phi^{(1)}) = \gamma' \mathbb{E}_{\substack{b^{(2)} = \phi^{(1)} \\ \phi^{(2)} \sim \pi^{(2)}(\cdot|b^{(2)})}} \left[ Q^{(2)}(b^{(2)}, \phi^{(2)}) \right]$$

In general, $\phi^{(1)} \in \Delta(A^{(1)})$ is continuous. However, this can be simplified if we only consider deterministic $\phi$:

$$Q^{(1)}(\delta_A^{(1)}) = \gamma' \mathbb{E}_{a^{(2)} \sim \pi^{(2)}(\cdot|b^{(2)}=A)} \left[ Q^{(2)}(b^{(2)} = A, a^{(2)}) \right]$$

$$Q^{(1)}(\delta_B^{(1)}) = \gamma' \mathbb{E}_{a^{(2)} \sim \pi^{(2)}(\cdot|b^{(2)}=B)} \left[ Q^{(2)}(b^{(2)} = B, a^{(2)}) \right]$$

$$Q^{(1)}(\delta_C^{(1)}) = \gamma' \mathbb{E}_{a^{(2)} \sim \pi^{(2)}(\cdot|b^{(2)}=C)} \left[ Q^{(2)}(b^{(2)} = C, a^{(2)}) \right]$$

where we have used $b^{(2)} = A$ to denote the fact that agent 2 knows with probability 1 that it is in state $A$ since it knows that $\pi^{(1)}$ chooses $A$ deterministically. Also, we used the notation $\delta_A^{(1)}$ to denote a particular action distribution $\phi$ which deterministically selects $A$.

For agent 2, policy evaluation is simply the reward function given in the Matrix game:

$$Q^{(2)}(b^{(2)} = A, A) = R(A, A) = 5$$

$$\vdots$$

$$Q^{(2)}(b^{(2)} = B, B) = R(B, B) = 10$$

$$\vdots$$

$$Q^{(2)}(b^{(2)} = C, C) = R(C, C) = 20$$

For policy improvement,

$$\pi^{(2)}(b^{(2)} = A) \leftarrow \arg\max_{a^{(2)} \in \{A,B,C\}} Q^{(2)}(b^{(2)} = A, a^{(2)})$$

$$\pi^{(2)}(b^{(2)} = B) \leftarrow \arg\max_{a^{(2)} \in \{A,B,C\}} Q^{(2)}(b^{(2)} = B, a^{(2)})$$

$$\pi^{(2)}(b^{(2)} = C) \leftarrow \arg\max_{a^{(2)} \in \{A,B,C\}} Q^{(2)}(b^{(2)} = C, a^{(2)})$$

.

Thus, $\pi^{(2)}$ will select $A, B, C$ given agent 1 deterministically selects $A, B, C$, respectively.

For agent 1,

$$\pi^{(1)} \leftarrow \arg\max_{a^{(1)} \in \{A,B,C\}} \gamma' Q^{(2)}(b^{(2)} = a^{(1)}, \pi^{(2)}(b^{(2)} = a^{(1)}))$$

Now, let's say we are in an adversarial starting point where the policy is initialized to deterministically select $(A, A)$.

Initially, policy improvement for $\pi^{(1)}$ leads agent 1 to continue selecting $A$ since $\pi^{(2)}$ deterministically selects A and $R(A, A) = 5$ is better than $R(B, A) = R(C, A) = -20$. However, after the first iteration, agent 2 will update its policy to select $A, B, C$ for $b^{(2)} = A, b^{(2)} = B, b^{(2)} = C$, respectively. Thus, agent 1 in iteration 2 will select $C$ since $C = \arg\max_{a^{(1)}} \gamma' Q^{(2)}(b^{(2)} = a^{(1)}, \pi^{(2)})$, where $\pi^{(2)}$ is now updated.

## H   LEARNING CURVE FOR SMACV2

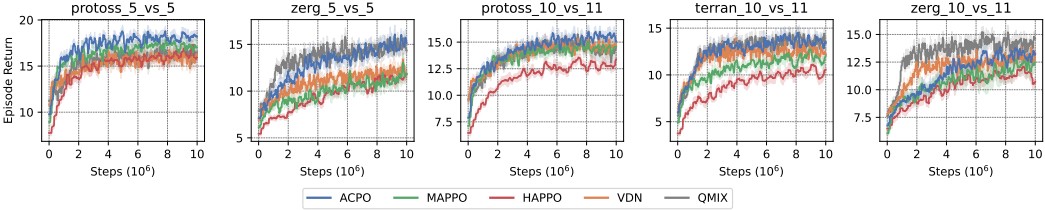

Figure 7: Return for SMACv2 with mean and standard error over 10 seeds.

## I   HYPERPARAMETER DETAILS

For a fair comparison, we set the network type (MLP or GRU) and hidden layer size to be consistent across all algorithms. The design choices follow the experimental setups of Zhong et al. (2024) and Papoudakis et al. (2021). The discount factor $\gamma$ is fixed, as it is inherent to the MMDP rather than a tunable hyperparameter. In contrast ACPO employs serialization, where the advantage is computed using $\gamma' = \gamma^{1/N}$.

For all baselines, we use the reported hyperparameters from Papoudakis et al. (2021) for RWARE, Ellis et al. (2023) for SMACv2 and Zhong et al. (2024) for MA-MuJoCo. The full set of hyperparameters are provided in our anonymous code at the following link: `https://anonymous.4open.science/r/anonymous-acpo-BD51`.

Table 3: Common Parameters for All Algorithms

| Parameter | RWARE | MA-MuJoCo | SMACv2 |
|---|---|---|---|
| Network Hidden Sizes | MLP [128, 256] | MLP [128, 128, 128] | GRU [64] |
| $\gamma$ | 0.99 | 0.99 | 0.99 |

## J  COMPUTATIONAL RESOURCES

For RWARE experiments, we utilized a single NVIDIA GeForce RTX 3090 graphics processing unit (GPU). For the 2,4,8, and 12-agent environments, ACPO took 7H, 6H, 8H and 9H, respectively. The corresponding times were MAPPO (6H, 4H, 4H, 3H), HAPPO (19H, 37H, 71H, 107H) and HATRPO (21H, 38H, 70H, 105H).

In SMACv2 experiments, the hardware configuration depended on the scenario: a single NVIDIA GeForce RTX 3090 was used for 5-agent scenarios, whereas a single NVIDIA L40S GPU was used for 10-agent scenarios. For the 5-agent and 10-agent scenarios, ACPO took 16H and 20H, respectively. QMIX completed training in 24H and 34H, HAPPO required 33H and 51H, and MAPPO's training times were 12H and 15H for the respective scenarios.

Due to significant computational requirements, experiments for HATRPO on SMACv2 were not conducted. We estimated that reaching 10M timesteps would take approximately 14 days for the 5-agent scenarios and 40 days for the 10-agent scenarios, which was deemed computationally prohibitive for our resources. Moreover, Zhong et al. (2024) reported weaker performance in SMACv2 compared to HAPPO, MAPPO, and QMIX.

## K  DISCUSSION ON CENTRALIZED TRAINING WITH DECENTRALIZED EXECUTION

In our work, we consider the standard MARL paradigm of Centralized Training Decentralized Execution (CTDE), which allows multiple policies to be jointly trained but must be executed in a decentralized fashion.

Here we introduce related work in three different settings that are often considered in MARL: Centralized Training Centralized Execution (CTCE), CTDE with Centralized Rollouts (CTDE-CR), and CTDE with Decentralized Rollouts (CTDE-DR).

**Centralized Training Centralized Execution (CTCE)**  Centralized Training Centralized Execution (CTCE) methods such as Multi-Agent Transformer (MAT) (Wen et al., 2022) use a joint policy of the form $\pi(a^{(1)}, \ldots, a^{(N)}|s)$ during both training rollouts and execution. MAT is a centralized Transformer model defined on the joint action space, and uses a joint observation encoder and joint policies during both training and execution. While a decentralized policy version is also considered, MAT requires a joint observation encoder during both training and execution. In MMDPs, the CTCE setting reduces to a Factored-Action MDP (Guestrin et al., 2001; Raghavan et al., 2012), which is a single-agent MDP with factored action spaces. In this case, single-agent techniques such as policy iteration and value iteration can be applied directly.

Generally, centralized control (CTCE) is not applicable to many real-world multi-agent systems such as power grids Wang et al. (2021a), traffic signal control Chu et al. (2020), and large-scale fleet management Lin et al. (2018) due to the large joint action space and prohibitive communication costs.

**Centralized Training Decentralized Execution with Centralized Rollouts (CTDE-CR)**  There are also some work within CTDE but with additional assumptions to make training simpler. (Ye et al., 2023; Wang et al., 2023) aim to decentralize centralized solutions Amato (2024), by assuming that a single-agent joint policy can be used for training rollout. This joint policy is used during centralized training and distilled to decentralized policies before execution. As a single-agent problem,

this assumption makes convergence to optimal policies straightforward as in the CTCE case. We can view this setting as a special case of our work where we assume access to a joint policy during training rollouts (CTDE-CR). With this additional assumption, we can solve the serialized problem introduced in Section 3 without considering beliefs. However, this line of research inherits similar weaknesses of CTCE, and cannot be applied to many real-world multi-agent systems with a massive action space or prohibitive communication costs.

**Centralized Training Decentralized Execution with Decentralized Rollouts (CTDE-DR)**   In CTDE-DR, the policy must be decentralized (fully factorized) during both training and execution, with the policy form $\vec{\pi} = \langle \pi^{(1)}, \dots \pi^{(N)} \rangle$ where $\pi^{(i)} : S \to A^{(i)}$. This is the natural MARL paradigm we consider in our work. Algorithms for simultaneous policy update methods Lowe et al. (2017); Yu et al. (2022), iterative best response methods Kuba et al. (2022); Zhong et al. (2024); Liu et al. (2024) as well as value decomposition methods Rashid et al. (2018); Zhang et al. (2021) all fall under CTDE-DR.

## L    BELIEF APPROXIMATION FOR PRACTICAL IMPLEMENTATIONS

In ACPO (PPO-based), the clipped policy gradient objective in Eq. 3 is similar to MAPPO with an additional input to the policy, the belief $b^{(i)}$. In high-dimensional domains, the belief update for a POMDP is intractable to compute exactly and is often approximated with RNNs or Transformers (Ni et al., 2022; 2023). For AC-BMDPs, the belief is defined as a distribution over the unobservable $\vec{a}^{<i}$, which we approximate by having each agent predict the previous agents' actions for the current time step.

Below we show our overall opponent modeling procedure using state inputs for simplicity[4].

First, Agent 1 has no notion of belief, so it simply outputs $a^{(1)} \sim \pi^{(1)}(\cdot \mid s)$. Next, agent 2 predicts the action that agent 1 has taken, i.e. $\tilde{a}^{(1)} \sim \tilde{\pi}^{(1)}(\cdot \mid s)$, where $\tilde{\pi}^{(1)}$ is agent 2's own policy with an additional agent ID of 1 as input. Now, agent 2 will output its action, i.e. $a^{(2)} \sim \pi^{(2)}(\cdot \mid s, \tilde{a}^{(1)})$ using the predicted action for agent 1.

## M    RETURN VS RUNTIME COMPARISON ON RWARE

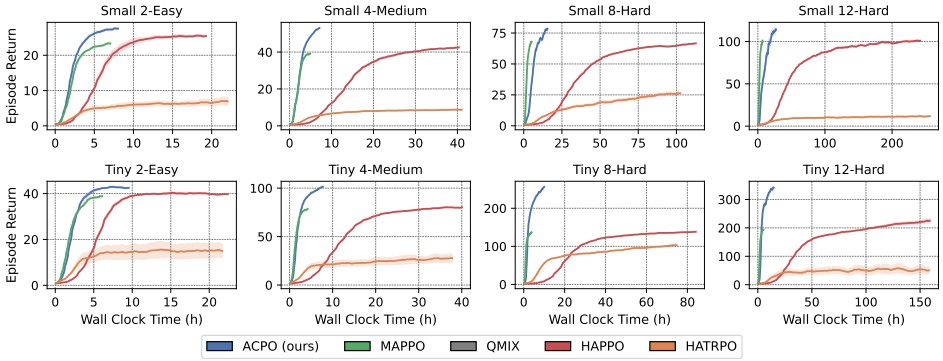

Figure 8: Episode return vs training runtime comparison on Multi-Robot Warehouse (RWARE). The mean and standard error over 10 seeds are reported for all tasks and algorithms except HATRPO and HAPPO on Small 12-Hard (5 seeds) and Tiny 12-Hard (8 seeds).

Figure 8 compares the episode returns as a function of the training runtime. The experiments were terminated once the maximum number of timesteps was reached, as depicted in Figure 2. Notably,

---

[4]For partially observable environments, we replace the state $s_t$ with either its own partial observation $o_t^{(i)}$ for MLPs or the hidden state of an RNN $h_t^{(i)}$ which takes the action-observation history $\langle o_{<t}^{(i)}, \vec{a}_t^{(i)} \rangle$ as input.

ACPO and MAPPO demonstrate substantially better runtime efficiency compared to HAPPO and HATRPO, primarily due to their simultaneous rather than sequential policy updates. Furthermore, ACPO consistently outperforms the other methods across all settings while maintaining a runtime comparable to MAPPO. The slight increase in ACPO's runtime is attributed to the additional cost of belief computation.

## N    AGENT-CHAINED TWIN DELAYED DETERMINISTIC POLICY GRADIENT (AC-TD3)

Following the Bellman operators in Definition 5.1, the critic loss based on TD3 (Fujimoto et al., 2018) can be written as follows:

for $i = 1, \ldots, N-1$,

$$J_Q^{(i)}(\psi) = \mathbb{E}_{\substack{(s, b^{(i)}, \phi^{(i)}) \sim \mathcal{D} \\ \phi^{(i+1)} \sim \pi_\theta^{(i+1)}(\cdot | s, b^{(i)}, \phi^{(i)})}} \left[ \left( Q_\psi^{(i)}([s, b^{(i)}], \phi^{(i)}) - y^{(i)} \right)^2 \right]$$

$$\text{s.t. } y^{(i)} = \gamma' Q_\psi^{(i+1)}([s, b^{(i)}, \phi^{(i)}], \phi^{(i+1)})$$

$$J_Q^{(N)}(\psi) = \mathbb{E}_{\substack{(s, b^{(N)}, \phi^{(N)}, r, s') \sim \mathcal{D} \\ \phi^{(1)'} \sim \pi_\theta^{(1)}(\cdot | s')}} \left[ \left( Q_\psi^{(N)}([s, b^{(N)}], \phi^{(N)}) - y^{(N)} \right)^2 \right]$$

$$\text{s.t. } y^{(N)} = r + \gamma' Q_\psi^{(1)}([s', \phi^{(1)'}])$$

We simplify this objective by considering deterministic $\phi^{(i)}$,

for $i = 1, \ldots, N-1$,

$$J_Q^{(i)}(\psi) = \mathbb{E}_{\substack{(s, \vec{a}^{<i}, a^{(i)}) \sim \mathcal{D} \\ a^{(i+1)} \sim \pi_\theta^{(i+1)}(\cdot | s, \vec{a}^{<i}, a^{(i)})}} \left[ \left( Q_\psi^{(i)}([s, \vec{a}^{<i}], a^{(i)}) - y^{(i)} \right)^2 \right]$$

$$\text{s.t. } y^{(i)} = \gamma' Q_\psi^{(i+1)}([s, \vec{a}^{<i}, a^{(i)}], a^{(i+1)})$$

$$J_Q^{(N)}(\psi) = \mathbb{E}_{\substack{(s, \vec{a}^{<N}, a^{(N)}, r, s') \sim \mathcal{D} \\ \vec{a}' \sim \vec{\pi}_\theta(\cdot | s')}} \left[ \left( Q_\psi^{(N)}([s, \vec{a}^{<N}], a^{(N)}) - y^{(N)} \right)^2 \right]$$

$$\text{s.t. } y^{(N)} = r + \gamma' Q_\psi^{(1)}([s', a^{(1)'}])$$

For practical implementations, it is often useful to consider $k$-step returns.

$$J_Q^{(i)}(\psi) = \mathbb{E}_{\substack{(s_t, \vec{a}_t^{<i}, a_t^{(i)}, \{r_{t+j}\}_{j=0}^k, s_{t+k+1}) \sim \mathcal{D} \\ \vec{a}_{t+k+1} \sim \vec{\pi}_\theta(\cdot | s_{t+k+1})}} \left[ \left( Q_\psi^{(i)}([s_t, \vec{a}_t^{<i}], a_t^{(i)}) - (\gamma')^{N-i} y_t^{(i)} \right)^2 \right]$$

$$\text{s.t. } y_t^{(i)} = r_t + \gamma r_{t+1} + \cdots + \gamma^k r_{t+k} + \gamma^{k+1} Q_\psi^{(i+1)}([s_{t+k+1}, \vec{a}_{t+k+1}^{<i}], a_{t+k+1}^{(i+1)})$$

We find that using $k$-step returns in this way works better in practice as each agent now has a dense reward signal in the targets (rather than only the last agent). We note that $\gamma$ denotes the discount factor in the original MMDP and $\gamma' = \gamma^{1/N}$. The $(\gamma')^{N-i}$ discount is to adjust the micro step to match with the last agent. For example, for agent 1, the reward given at the current timestep is $(\gamma')^{N-1} r_t$.

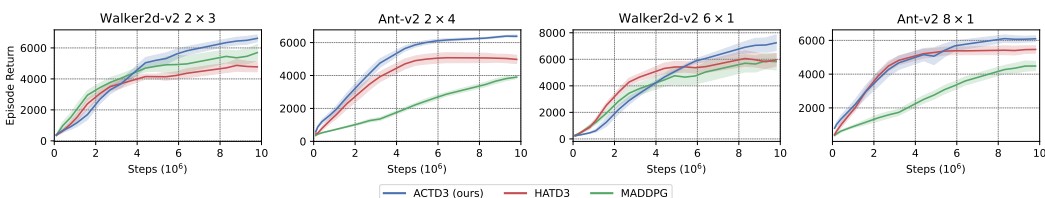

Figure 9: Mean Return and Standard Error over 5 seeds for MA-MuJoCo (Gym).

## O  OFF-POLICY COMPARISON

We evaluate an off-policy variant of ACPO, termed AC-TD3, by incorporating agent chaining into the TD3 (Fujimoto et al., 2018) algorithm, as described in Appendix J. The performance of ACTD3 is compared against off-policy baselines, HATD3 (Zhong et al., 2024) and MADDPG (Lowe et al., 2017). Experiments are conducted on the Ma-MuJoCo (Gym) environment using five random seeds. As illustrated in Figure 9, AC-TD3 consistently outperforms all baseline methods.

