# OpenReview forum: "Agent-Chained Policy Optimization"
_ICLR.cc/2026/Conference — Submitted to ICLR 2026_

### Official Review · Reviewer_ScUU · 2025-10-29

**Soundness:** 2
**Presentation:** 3
**Contribution:** 3
**Rating:** 2
**Confidence:** 4

**Summary:**

This works introduces a paradigm for centralised training of actor critic algorithms in multi-agent settings which allows agent-specific value functions to incorporate beliefs over the actions taken by other agents. The authors introduce a formulation for simultaneous actions taken by agents with sequential belief chaining, and prove that it converges to a globally optimal policy.

**Strengths:**

1. The work is well motivated, as the myopic nature of simultaneous updates in MARL results in inconsistent updates between agents. ACPO is a natural framework to subvert this issue while retaining the benefits of centralised training.
2. The proofs appear correct. (See questions)
3. The authors cover a good selection of benchmarks with continuous and discrete control.
4. The didactic game in Table 1 is easy to follow and a useful example. Figure 1 is also a good explanation for the concept of belief chaining.

**Weaknesses:**

1. The authors implement ACPO with PPO , but do not indicate whether their method can be extended to Soft Actor Critic[1] or other common actor-critic algorithms. Why was PPO chosen solely? The authors must be transparent about whether this choice was due to implementation difficulties or a lack of performance generalisation.
2. Limited baselines - the authors compare to MAPPO (general MARL AC), HAPPO and HATRPO (BR MARL AC) and QMIX in smacv2. Why were value based methods and more actor critic methods not compared against in these benchmarks? There are existing implementations of VDN, MADDPG, COMIX, and other standard MARL algorithms for the benchmarks provided - if no relevant BR baselines are available, it is at least necessary to understand how ACPO with PPO compares to common MARL baselines. 3. The computational comparison in Figure 5 might be a little unfair, as the BR methods dominate the plot - it is hard to notice that for the 12-hard scenario, ACPO takes seemingly 3-4 times as much as MAPPO to solve the task, despite not reaching a similar multiplier in performance.
4. Why were the main results obtained with only 3 seeds? Computational constraints are a reasonable answer, but a standard of 5-10 seeds should be the minimum for these results to be considered reliable.

**Questions:**

1. Can this work for non-cooperative settings? What about mixed objective games?
2. Why does Theorem C.1. use a deterministic policy? Why can a stochastic policy not be used here?
3. Why are the computational comparisons between ACPO and MAPPO so unfavourable? Have the authors missed an implementation detail that could lead to faster runtimes?
4. Must the agent order always remain the same? What if it were randomized? Does the current ordering not result in asymmetric learning dynamics amongst the agents, even in symmetric games?

---

> ### Author Response · Authors · 2025-11-21
>
> # Response to Reviewer ScUU (1/3)
>
> > **The authors implement ACPO with PPO , but do not indicate whether their method can be extended to Soft Actor Critic[1] or other common actor-critic algorithms. Why was PPO chosen solely? The authors must be transparent about whether this choice was due to implementation difficulties or a lack of performance generalisation.**
>
> The main focus of our experiments was to evaluate the effect of agent-chaining over state-of-the-art methods in simultaneous policy update (MAPPO) and iterative best response (HAPPO). Since both methods share the same PPO backbone, we can isolate the effect of agent-chaining by building ACPO on top of PPO as well. Furthermore, PPO and MAPPO are known to have stable training dynamics in both single-agent and multi-agent RL, and generally perform well across a wide range of environments.
>
> However, ACPO is a general algorithm-agnostic method which can be applied to common actor-critic methods as well. To demonstrate this, we have implemented Agent-Chained TD3 (ACTD3) which approximates ACPI with Twin Delayed Deterministic Policy Gradients (TD3) (Fujimoto et al., 2019) an off-policy actor-critic method which learns a Q-value as the critic. We follow Heterogenous TD3 (HATD3) (Zhong et al., 2024) for the implementation of ACTD3. The main difference is that ACTD3 learns decentralized critics (Q networks) for each agent while HATD3 uses a centralized critic and applies iterative best response similar to HAPPO and HATRPO.
>
> | Environment | MADDPG | HATD3 | ACTD3 |
> |-|-|-|-|
> | Walker2d-v2_2x3 | 5745.85 ± 552.65 | 4738.43 ± 297.91 | **6629.08 ± 215.01** |
> | Ant-v2_2x4 | 3903.73 ± 98.22 | 4954.84 ± 250.42 | **6400.60 ± 91.42** |
>
> Table A.1: Mean return and standard error over 5 seeds on MA-MuJoCo (Gym)
>
> We observe similar results with our on-policy experiments, where ACTD3 outperforms standard off-policy baseline including MADDPG and HATD3, using the same hyperparameters as HATD3. The performance gap is higher for Ant  which is a more challenging task with higher-dimensions in comparison to Walker2d.
>
>
> We provide details on ACTD3 in Appendix N and the learning curves in Appendix O.
>
>
> > **Limited baselines - the authors compare to MAPPO (general MARL AC), HAPPO and HATRPO (BR MARL AC) and QMIX in smacv2. Why were value based methods and more actor critic methods not compared against in these benchmarks? There are existing implementations of VDN, MADDPG, COMIX, and other standard MARL algorithms for the benchmarks provided - if no relevant BR baselines are available, it is at least necessary to understand how ACPO with PPO compares to common MARL baselines.**
>
>
> Please see Table 2 (SMACv2) in the updated manuscript and the Table A.1 (Gym MA-MuJoCo) from the previous question where we have added VDN (Sunehag et al., 2018) and MADDPG (Lowe et al., 2017) as additional baselines. The results support the analysis from Zhong et al., 2024 which also showed MADDPG is a weaker baseline compared to the more recent methods we have considered.
> VDN is also outperformed by QMIX in all tasks as shown in Table 2, and further supports the claim that VDN (linear combination of individual utility functions) is a more restrictive method of value decomposition compared to QMIX (weighted combination of individual utility functions). Please also see the response to Reviewer EQRP for details on the limitations of value decomposition.
>
> As we mention in Section 7, our main baselines we consider are MAPPO and HAPPO which are the strongest on-policy baselines in the environments we consider, and are representative methods for simultaneous policy update and iterative best response, respectively.
> We have also added QMIX as an off-policy baseline since it has shown strong performance on SMACv2, but does not learn any meaningful behavior in Multi-Robot Warehouse.
> We show that ACPO outperforms MAPPO and HAPPO using the same hyperparameters as MAPPO, and demonstrate the positive effect of agent-chaining, which is the primary goal of our experiments.
>
> We do not compare against COMIX (Minelli et al., 2024) as it explicitly tackles the effect of  communication in MARL and considers a different set of benchmarks, environmental setups and baselines. In fact, COMIX without the explicit communication channel reduces to QMIX. In principle, similar communication channels can be added to ACPO, and we consider this to be an interesting avenue for future work.
>
> Finally, we note that there are no standard implementations of COMIX on the benchmarks we consider as it considers an entirely different set of benchmarks.

---

> ### Author Response · Authors · 2025-11-21
>
> # Response to Reviewer ScUU (2/3)
>
> > **Why are the computational comparisons between ACPO and MAPPO so unfavourable? Have the authors missed an implementation detail that could lead to faster runtimes?**
>
> > **3. The computational comparison in Figure 5 might be a little unfair, as the BR methods dominate the plot - it is hard to notice that for the 12-hard scenario, ACPO takes seemingly 3-4 times as much as MAPPO to solve the task, despite not reaching a similar multiplier in performance.**
>
> The additional overhead compared to MAPPO comes primarily from opponent modeling where each agent $i$ must predict the actions of the preceding agents $1, \dots i-1$ (details on opponent modeling for ACPO are provided in detail in Appendix L).
> The goal of Figure 5 is to show that this overhead is minimal compared to the computational bottleneck from sequential policy updates in iterative best response methods such as HAPPO/HATRPO. The total training time for HAPPO and HATRPO increases linearly where it takes approximately 2 hours for 2 agents and 12 hours for 12 agents for 5M timesteps. For ACPO, the total training time remains within 1-2 hours for both 2 and 12 agents. We believe this additional overhead is justified since ACPO outperforms baselines with a substantial gap especially when the number of agents increase and higher levels of coordination are required.
>
> For clarity, we provide the return curve for Multi-Robot Warehouse (RWARE) where the x-axis is the wall-clock time in hours (Figure 8 in Appendix M). We see that MAPPO and ACPO always finish training at a much earlier time than HAPPO and HATRPO. The return for ACPO is substantially higher than all baselines including MAPPO, especially as the number of agents increase.
>
> Finally, we respectfully disagree that there must be a direct linear relationship between runtime and return. The performance is bounded by the optimal policy of the environment, and it becomes exponentially harder to improve performance as the return approaches that of the optimal policy. The standard protocol in online RL is to primarily compare algorithms based on sample efficiency (the number of environment interaction steps required for achieving a certain level of return). Wall-clock training time is an auxiliary metric we provide to consider the practical implications of longer training time as in HAPPO and HATRPO.
>
> > **Why were the main results obtained with only 3 seeds? Computational constraints are a reasonable answer, but a standard of 5-10 seeds should be the minimum for these results to be considered reliable.**
>
> We have updated Figure 2 (RWARE), Figure 3 (MA-MuJoCo), Table 2 (SMACv2) and Figure 7 (SMACv2) in our manuscript. The updated results are for **10 seeds** on RWARE, SMACv2, and MA-MuJoCo. There is no significant change in the empirical results, and our analysis and conclusions remain the same.
>
> We initially followed the experimental protocol in HARL (Zhong et al., 2024) which also ran 3 seeds to ensure a wide coverage of environments, tasks and baselines, as well as statistical significance.
>
> (Note) The additional seeds for 8 and 12 agent tasks for RWARE will be added later during the discussion phase, as it takes considerably longer training time for the best response baselines (HAPPO and HATRPO).

---

> ### Author Response · Authors · 2025-11-21
>
> # Response to Reviewer ScUU (3/3)
>
> > **Can this work for non-cooperative settings? What about mixed objective games?**
>
> We primarily focus on the cooperative setting and address the long-standing challenge of going beyond Nash Equilibrium and targeting convergence to the globally optimal policy under CTDE. This follows a long line of work such as QMIX (Rashid et al., 2018), MAPPO (Yu et al., 2022), HAPPO (Zhong et al., 2024) and many more which focused on the cooperative setting where the goal is to learn coordination behavior under decentralized control.
>
> In non-cooperative settings where each agent has different reward functions, the notion of optimality is defined by a particular choice of equilibria, such as Nash Equilibrium, Correlated Equilibrium, Quantal Response Equilibrium, Pareto-optimality, etc. To the best of our knowledge, there is no notion of *global optimality* that the research community agrees upon. Thus, extending our work and deriving a theoretically grounded approach for the non-cooperative setting is not straightforward.
>
> Nonetheless, further exploring the ideas in our work to the non-cooperative setting is an interesting avenue for future work.
>
>
>
> > **Why does Theorem C.1. use a deterministic policy? Why can a stochastic policy not be used here?**
>
> As detailed in Appendix E, Algorithm 1 provides a straightforward policy iteration procedure defined on the AC-BMDP.  However, the space of action distributions $\phi^{(i)}$ is infinite, which makes this intractable even in tabular domains. A direct corollary of Theorem C.1 is that it is sufficient to consider a finite action space. This allows us to define a tractable policy iteration procedure (Algorithm 2), as well as provide a formal connection between the AC-BMDP and the underlying MMDP.
>
> We also note that the consideration of deterministic policies is for theoretical purposes. ACPI follows policy iteration which ensures convergence to an optimal deterministic policy, which exists for every MDP. In practice, our practical implementation (ACPO) uses stochastic policies (e.g., Gaussians, Categorical) to facilitate gradient-based optimization in high-dimensional spaces. This follows the standard paradigm of algorithms like PPO, where the stochastic policy can asymptotically approach the optimal deterministic policy (e.g., as variance approaches 0).
>
>
> > **Must the agent order always remain the same? What if it were randomized? Does the current ordering not result in asymmetric learning dynamics amongst the agents, even in symmetric games?**
>
>
> The agent order can be randomized or domain knowledge can be used to consider specific orders. However, once chosen, this order should be fixed throughout training to ensure the policy inputs remain consistent.
>
> In terms of asymmetric learning dynamics, we agree that the learning dynamics can inherently be asymmetric due to different agents having a different space of beliefs. However, our empirical results show that this does not hurt performance.
> Since we cover a wide range of environments in our experimental results, the degree of symmetry is also diverse.  ACPO is robust to seeds and outperforms baselines in terms of sample complexity and asymptotic performance, especially as the number of agents and degree of coordination required increases.
>
> Overall, our goal in this work is to tackle general cooperative MARL settings where decentralized execution remains a challenge.

---

> ### Author Response · Authors · 2025-11-21
>
> ## References
>
> Fujimoto, Scott, Herke van Hoof, and David Meger. "Addressing Function Approximation Error in Actor-Critic Methods." International Conference on Machine Learning (ICML) (2018).
>
> Lowe, Ryan, Yi Wu, Aviv Tamar, Jean Harb, OpenAI Pieter Abbeel, and Igor Mordatch. “Multi-Agent Actor-Critic for Mixed Cooperative-Competitive Environments.” NeurIPS (2017).
>
> Minelli, Giovanni, and Mirco Musolesi. “CoMIX: A Multi-agent Reinforcement Learning Training Architecture for Efficient Decentralized Coordination and Independent Decision-Making.” Transactions on Machine Learning Research (TMLR) (2024).
>
> Rashid, Tabish, Mikayel Samvelyan, Christian Schroeder de Witt, Gregory Farquhar, Jakob Foerster, and Shimon Whiteson. “QMIX: Monotonic Value Function Factorisation for Deep Multi-Agent Reinforcement Learning.” ICML (2018).
>
> Sunehag, Peter, Guy Lever, Audrunas Gruslys, Wojciech Marian Czarnecki, Vinicius Zambaldi, Max Jaderberg, Marc Lanctot, Nicolas Sonnerat, Joel Z. Leibo, Karl Tuyls, and Thore Graepel. “Value-Decomposition Networks For Cooperative Multi-Agent Learning Based On Team Reward.” AAMAS (2018).
>
> Yu, Chao, Akash Velu, Eugene Vinitsky, Jiaxuan Gao, Yu Wang, Alexandre Bayen, and Yi Wu. “The Surprising Effectiveness of PPO in Cooperative, Multi-Agent Games.” NeurIPS Datasets and Benchmarks Track (2022)
>
> Zhong, Yifan, Jakub Grudzien Kuba, Xidong Feng, Siyi Hu, Jiaming Ji, and Yaodong Yang. “Heterogeneous-Agent Reinforcement Learning.” Journal of Machine Learning Research (JMLR) (2024).

---

> ### Author Response · Authors · 2025-11-27
>
> Dear Reviewer ScUU,
>
> Thank you again for taking the time to review our work. Since the initial reviews, we have updated the manuscript with:
>
> * Results for 10 seeds (previously 3) on RWARE, SMACv2, MA‑MuJoCo (Figure 2-3, Table 2, Figure 7) with conclusions unchanged
> * Algorithm Details and Results for Agent-Chained TD3 (ACTD3) a new off-policy variant of our algorithm (Appendix N-O)
> * Additional baselines: VDN, MADDPG, HATD3 (Table 2, Figure 7, Figure 9)
> * Detailed analysis and discussion on runtime statistics (Appendix M)
>
> We would be very grateful if you could let us know if there are any remaining major concerns after these updates, or if the current version sufficiently addresses the earlier weaknesses that were raised. This would greatly help us understand how to further improve the work, and address any remaining concerns if any.

---

### Official Review · Reviewer_EQRP · 2025-10-30

**Soundness:** 4
**Presentation:** 4
**Contribution:** 3
**Rating:** 4
**Confidence:** 3

**Summary:**

The paper introduces a new framework for Cooperative Multi-Agent Reinforcement Learning that guarantees convergence to the globally optimal joint policy, overcoming limitations of existing methods that either converge to local Nash equilibria or rely on centralized critics without convergence guarantees.

**Strengths:**

Originality:

The idea of serializing multi-agent decision making via a belief MDP that maintains consistency with the decentralized setting is novel.
The agent-chained construction elegantly bridges the gap between centralized critics and decentralized value functions, creating a unified formulation that retains theoretical guarantees without restrictive assumptions.
Proving convergence to the global optimum rather than Nash equilibrium represents a significant conceptual advance over HAPPO, HATRPO, and MAPPO.

Quality:

The paper is well-grounded theoretically, with rigorous derivations and clear definitions.
The ablation study and runtime comparison are particularly convincing, showing that performance gains stem from the agent-chaining mechanism rather than auxiliary factors.

Clarity:


Mathematical notation is consistent and readable.
Proof sketches are presented intuitively, with full details deferred to appendices.

Significance:

Theoretically, the paper closes a long-standing gap in cooperative MARL by providing a convergence-guaranteed formulation under the widely-used CTDE paradigm.
Practically, ACPO shows robust scalability as the number of agents increases, which has been a critical bottleneck in MARL.

**Weaknesses:**

The AC-BMDP belief update assumes tractable distributions over prior agents’ actions. In practice, belief approximation or parameterization details are underspecified, and real-world scalability of belief updates may be challenging.

No comparison to modern off-policy or model-based MARL methods. Although justified for fairness, this limits understanding of ACPO’s off-policy robustness.


The ablation only removes agent-chaining; finer-grained tests could clarify which theoretical component drives the performance gain.

**Questions:**

The proofs assume deterministic policies; how robust is ACPI’s convergence under stochastic policies used in ACPO?

Could the agent-chained structure be combined with value-decomposition methods to handle credit assignment more efficiently?

The authors mention applications to Multi-Agent LLMs. Could the same belief over prior actions mechanism be re-interpreted as a reasoning chain in communication-based coordination tasks?

---

> ### Author Response · Authors · 2025-11-21
>
> # Response to Reviewer  EQRP (1/2)
> > **The AC-BMDP belief update assumes tractable distributions over prior agents’ actions. In practice, belief approximation or parameterization details are underspecified, and real-world scalability of belief updates may be challenging.**
>
> In high-dimensional domains, the belief update for a POMDP is intractable to compute exactly and is often approximated by feeding the history to an RNN or Transformer (Ni et al., 2022, 2023). We also do not compute the belief explicitly, but instead approximate this with opponent modeling.
> In particular, for AC-BMDPs, the belief is defined as a distribution over the unobservable $\vec{a}^{<i} $, which we approximate by having each agent predict the previous agents' actions for the current time step.
>
> Below we show our overall opponent modeling procedure using state inputs for simplicity.
>
> First, agent 1 has no notion of belief, so it simply outputs
> $a^{(1)} \sim \pi^{(1)}(\cdot \mid s)$. For agent 2, it  predicts the action that agent 1 has taken, i.e. $\tilde{a}^{(1)} \sim \tilde{\pi}^{(1)}(\cdot \mid s)$, where $\tilde{\pi}^{(1)}$ is agent 2’s own policy with an additional agent ID of $1$ as input. Now, agent 2 will output its action, i.e. $a^{(2)} \sim \pi^{(2)}(\cdot \mid s, \tilde{a}^{(1)})$ using the predicted action for agent $1$.
>
> Finally, as mentioned in Section 4,  we note that predicting the actions taken by other agents has a rich history in MARL. However, unlike other work which uses opponent modeling as an auxiliary task, we derive this formally from the AC-BMDP and connect the opponent modeling module as computing a belief.
>
> > **No comparison to modern off-policy or model-based MARL methods. Although justified for fairness, this limits understanding of ACPO’s off-policy robustness.**
>
> Please see Figure 9 in Appendix O where we have provided additional results for Agent-Chained Twin Delayed Deep Deterministic Policy Gradients (ACTD3). ACTD3 is derived with a popular off-policy method TD3 (Fujimoto et al., 2018) on top of our agent-chaining framework. We show that ACTD3 is a general algorithm-agnostic method which can be applied to common (both on and off-policy) actor-critic methods including TD3. ACTD3 outperforms standard off-policy baselines including MADDPG (Lowe et al., 2017) and HATD3 (Zhong et al., 2024). The performance gap is wider for Ant which is a more challenging task with higher-dimensions in comparison to Walker2d.
>
> > **The ablation only removes agent-chaining; finer-grained tests could clarify which theoretical component drives the performance gain.**
>
> We emphasize that ACPO w/o Agent Chaining (provided in Figure 4) reduces to MAPPO with an additional belief as input. While the performance slightly improves over MAPPO on a few tasks, the gain is minimal, and we see that agent-chaining is the core component driving performance. This is based on our controlled experimental setup where ACPO uses the same hyperparameters as MAPPO in order to isolate the effect of agent-chaining.
>
> Please let us know if there are other specific ablation studies you would like us to consider.

---

> ### Author Response · Authors · 2025-11-21
>
> # Response to Reviewer  EQRP (2/2)
> > **The proofs assume deterministic policies; how robust is ACPI’s convergence under stochastic policies used in ACPO?**
>
> As AC-BMDPs are a single-agent MDP defined over the belief state space, the same properties for policy iteration in single-agent RL apply for ACPI.
>
> Our theoretical results for ACPI follow policy iteration, which is a canonical dynamic programming approach to solving MDPs. In tabular single-agent RL, policy iteration guarantees convergence to the optimal deterministic policy. This forms a foundation for many practical algorithms such as SAC and PPO.
> Since every MDP has an optimal deterministic policy, there is no need to search the infinite space of stochastic policies during policy iteration. While our theoretical proofs (ACPI) rely on this property, our practical implementation (ACPO) uses stochastic policies (e.g., Gaussians, Categorical) to facilitate gradient-based optimization in high-dimensional spaces. This follows the standard paradigm of algorithms like PPO, where the stochastic policy can asymptotically approach the optimal deterministic policy (e.g., as variance approaches 0).
>
>
> > **Could the agent-chained structure be combined with value-decomposition methods to handle credit assignment more efficiently?**
>
> In short, combining value decomposition with our framework is possible in principle but not desirable.
>
> Value decomposition assumes that the underlying value functions $Q^{global}$, can decompose into individual utility functions $q^{(i)}$. This line of research requires the Individual Global Max (IGM) assumption, which states that taking the greedy individual action with respect to the utility functions will result in the greedy joint action, i.e. $\arg \max_{a} Q^{global}(s, a) = (\arg \max_{ a^{(i)} } q^{(i)} (s, a^{(i)} ))_{i=1}^N$. However, IGM implicitly assumes that the optimal $Q^{*}$ is decomposable, which limits the space of MARL environments that value decomposition can handle. For instance, IGM is not satisfied for the XOR game which is a 2-player 2-action game which gives positive reward if the 2 agents choose different actions and 0 otherwise  (Fu et al., 2022).  Please also see 9.5.1 in Albrecht et al., 2024 for details on IGM, and other work which explores the theoretical limitations of value decomposition (Wang et al., 2021 and Dou et al., 2022).
>
> In our work, we do not make any such assumptions and prove convergence to the global optimum in any MARL environment (for tabular domains). We note that for previous work, convergence to the global optimum is not guaranteed even for tabular domains unless the game is *decomposable*.
> Finally, ACPI naturally defines agent-specific value functions without any decomposition assumptions. Credit assignment is naturally handled through chained value functions.
>
>
>
> > **The authors mention applications to Multi-Agent LLMs. Could the same belief over prior actions mechanism be re-interpreted as a reasoning chain in communication-based coordination tasks?**
>
> Yes, many recent LLMs are reasoning-based, where the action space is a sequence of reasoning steps followed by the final output. If the reasoning steps are hidden from the other agents in an Multi-Agent LLM system (which is desirable for reducing the context window), this is analogous to the other agents’ actions being unobservable. We leave the extension of ACPO to Multi-Agent LLM for future work.

---

> ### Author Response · Authors · 2025-11-21
>
> ## References
>
> Albrecht, Stefano V., Filippos Christianos, and Lukas Schäfer. Multi-Agent Reinforcement Learning: Foundations and Modern Approaches. MIT Press (2024).​
>
> Dou, Zehao, Jakub Grudzien Kuba, and Yaodong Yang. “Understanding Value Decomposition Algorithms in Deep Cooperative Multi-Agent Reinforcement Learning.” arXiv preprint arXiv:2202.04868 (2022).​
>
>
> Fu, Wei, Chao Yu, Zelai Xu, Jiaqi Yang, and Yi Wu. “Revisiting Some Common Practices in Cooperative Multi-Agent Reinforcement Learning.” ICML (2022).
>
> Fujimoto, Scott, Herke Hoof, and David Meger. “Addressing Function Approximation Error in Actor-Critic Methods.” ICML (2018).
>
> Lowe, Ryan, Yi Wu, Aviv Tamar, Jean Harb, OpenAI Pieter Abbeel, and Igor Mordatch. “Multi-Agent Actor-Critic for Mixed Cooperative-Competitive Environments.” NeurIPS (2017).
>
>
> Ni, Tianwei, Benjamin Eysenbach, Sergey Levine, and Ruslan Salakhutdinov. “Recurrent Model-Free RL is a Strong Baseline for Many POMDPs.” ICML (2022)
>
> Ni, Tianwei, Michel Ma, Benjamin Eysenbach, and Pierre-Luc Bacon. “When Do Transformers Shine in RL? Decoupling Memory from Credit Assignment.” NeurIPS (2023).​
>
>
>
> Wang, Jianhao, Zhizhou Ren, Beining Han, Jianing Ye, and Chongjie Zhang. “Towards Understanding Cooperative Multi-Agent Q-Learning with Value Factorization.” NeurIPS (2021).​
>
> Zhong, Yifan, Jakub Grudzien Kuba, Xidong Feng, Siyi Hu, Jiaming Ji, and Yaodong Yang. “Heterogeneous-Agent Reinforcement Learning.” Journal of Machine Learning Research (JMLR) (2024).

---

> ### Author Response · Authors · 2025-11-27
>
> Dear Reviewer EQRP,
>
> Thank you again for taking the time to review our work. Since the initial reviews, we have updated the manuscript with:
>
> * Algorithm Details and Results for Agent-Chained TD3 (ACTD3) a new off-policy variant of our algorithm (Appendix N-O)
> * Additional baselines: VDN, MADDPG, HATD3 (Table 2, Figure 7, Figure 9)
> * Added Details on Belief Approximation for Practical Implementations (Appendix L)
>
> We would be very grateful if you could let us know if there are any remaining major concerns after these updates, or if the current version sufficiently addresses the earlier weaknesses that were raised. This would greatly help us understand how to further improve the work, and address any remaining concerns if any.

---

### Official Review · Reviewer_hsU2 · 2025-11-01

**Soundness:** 3
**Presentation:** 3
**Contribution:** 1
**Rating:** 0
**Confidence:** 4

**Summary:**

The paper introduces Agent-Chained Policy Optimization (ACPO), a MARL framework based on sequentializing joint actions into micro-steps. It formalizes the process as an Agent-Chained Belief MDP (AC-BMDP), where each agent maintains a belief over preceding agents' actions. A corresponding Agent-Chained Policy Iteration (ACPI) is proved to converge to the globally optimal joint policy. The practical implementation, ACPO, extends PPO with chain-based advantage estimation under CTDE. Experiments on RWARE, SMACv2, and MA-MuJoCo show consistent improvements over MAPPO and HAPPO.

**Strengths:**

- The paper is well-written and well-structured.

**Weaknesses:**

The paper’s literature review is insufficient. Sequential decision-making formulations in MARL have already been extensively explored, such as [1-3]. The proposed formulation and algorithm are largely a subset of [3], which already formalized the transformation of an MMDP into a sequential single-agent problem, discussed its global optimality, and applied it to PPO for MARL tasks. Moreover, the paper’s discussion of prior methods and their limitations is less detailed and rigorous than in [3].

[1] Bertsekas, Dimitri P. “Multiagent Rollout Algorithms and Reinforcement Learning.” arXiv preprint arXiv:1910.00120 (2019).

[2] Wen, Muning, Jakub Grudzien Kuba, Runji Lin, Weinan Zhang, Ying Wen, J. Wang, and Yaodong Yang. “Multi-Agent Reinforcement Learning is a Sequence Modeling Problem.” arXiv preprint arXiv:2205.14953 (2022).

[3] Ye, Jianing, Chenghao Li, Jianhao Wang, and Chongjie Zhang. “Towards Global Optimality in Cooperative MARL with the Transformation and Distillation Framework.” (2022).

**Questions:**

Please discuss the differences between the proposed ACPO and the prior work [1-3].

---

> ### Author Response · Authors · 2025-11-21
>
> # Response to Reviewer hsU2
>
> >**The paper’s literature review is insufficient. Sequential decision-making formulations in MARL have already been extensively explored, such as [1-3]. The proposed formulation and algorithm are largely a subset of [3], which already formalized the transformation of an MMDP into a sequential single-agent problem, discussed its global optimality, and applied it to PPO for MARL tasks. Moreover, the paper’s discussion of prior methods and their limitations is less detailed and rigorous than in [3].**
>
> In order to clarify the difference between the references mentioned by the reviewer and our work, we first provide the following definitions:
>
> 1. *Centralized Training Centralized Execution (CTCE)*: A  *joint policy* of the form $\pi(a^{(1)}, \dots ,a^{(N)}|s)$ is used during both training rollouts and execution.
>
> 1. *Centralized Training Decentralized Execution with Decentralized Rollouts (CTDE-DR)*: The policy must be decentralized (fully factorized) during both training and execution, with the policy form $\vec{\pi} = \langle \pi^{(1)}, \dots \pi^{(N)} \rangle$ where $\pi^{(i)}: S\rightarrow A^{(i)}$.
>
> 1. *Centralized Training Decentralized Execution with Centralized Rollouts (CTDE-CR)*: CTDE with the additional assumption that a joint policy can be used during training rollouts, but during execution, the policy must be decentralized (fully factorized).
>
> Generally, centralized control (CTCE or CTDE-CR) is not applicable to many real-world multi-agent systems due to the large joint action space and prohibitive communication costs.
> Examples include power grids (Wang et al., 2021), traffic signal control (Chu et al., 2019), and large-scale fleet management (Lin et al., 2018).
>
> Below, we contextualize each of the references within these 3 settings, and explain how they differ from our work. In short, our work considers CTDE-DR while the previous work mentioned by the reviewer considers CTCE or CTDE-CR.
>
> > [Ye et al., Arxiv (2022)]
>
> This work falls under CTDE-CR, while our work specifically considers the CTDE-DR setting. Thus, we respectfully disagree with the reviewer regarding our work being a subset of this work.
>
> Ye et al., 2022 introduces Transformation and Distillation (TAD) which solves a single-agent RL problem during training with a joint policy, and distills it to factorized policies. By assuming access to a single-agent (teacher) policy over the joint action space $\pi(a^{(1)}, \cdots  a^{(N)} \mid s)$, the problem becomes a single-agent MDP as in CTCE, and standard techniques such as dynamic programming (for theory in tabular cases) or PPO (for high-dimensional domains) can be directly applied without changes.
>
> In fact, we can view this setting as a special case of our work where we assume access to a joint policy during training rollouts (CTDE-CR). With this additional assumption considered in TAD, we can solve the serialized problem introduced in Section 3 of our paper without considering beliefs. While our algorithm derived under CTDE-DR can be applied to CTDE-CR, the reverse does not hold.
>
> > [Wen et al., NeurIPS (2022)]
>
> This work considers CTCE, where joint policies are represented by autoregressive policies of the form ${\pi}(a^{(1)}, \dots , a^{(N)} \mid s) = \pi^{(1) } (a^{(1) } \mid s) \pi^{(2) } (a^{(2) } \mid s, a^{(1)}) \cdots \pi^{(N)} ( a^{(N)} \mid a^{(1)}, \dots , a^{(N-1)})$. For CTCE with autoregressive policies allowed, the problem reduces to a single-agent Factored-Action MDP.
>
> > [Bertsekas Arxiv (2019)]
>
> This work is primarily an online planning method for centralized control, where the introduction of serialization is analogous to  Kovarik et al., 2023 and Peralez et al., 2025 (referenced in Section 3), which show equivalence between the MMDP and the serialized problem.
>
> As we have stated in the beginning of Section 3, serialization is a useful technique to view the underlying multi-agent problem as a single-agent one, but has yet to be applied in the Deep MARL setting under CTDE-DR. In particular, the serialized MDP considered in Bertsekas (2019) is only applicable to CTCE settings, where joint policies are allowed during execution. Conversely, we consider CTDE where decentralized (fully factorized) policies are required during both training rollouts and execution.
> We have added Bertsekas 2019 as an additional reference in Section 3.
>
>
> Overall, assuming access to a joint policy for rollout during training (CTCE and CTDE-CR) significantly simplifies the problem since it is a single-agent MDP with a factored action space. The core (open) challenge in MARL that we tackle in our work is learning decentralized (fully factorized) policies which maximize return and converging to the global optimum in CTDE-DR.
>
>
> Please also see Appendix K of the updated manuscript where we have provided a detailed discussion between our work and the references mentioned.

---

> ### Author Response · Authors · 2025-11-21
>
> ## References
>
> Bertsekas, Dimitri P. “Multiagent Rollout Algorithms and Reinforcement Learning.” arXiv preprint arXiv:1910.00120 (2019).​
>
> Chu, Tianshu, Jie Wang, Lara Codecà, and Zhaojian Li. “Multi-Agent Deep Reinforcement Learning for Large-scale Traffic Signal Control.” IEEE Transactions on Intelligent Transportation Systems (2019).​
>
> Lin, Kaixiang, Renyu Zhao, Zhe Xu, and Jiayu Zhou. “Efficient Large-Scale Fleet Management via Multi-Agent Deep Reinforcement Learning.” KDD (2018).​
>
> Lowe, Ryan, Yi Wu, Aviv Tamar, Jean Harb, OpenAI Pieter Abbeel, and Igor Mordatch. “Multi-Agent Actor-Critic for Mixed Cooperative-Competitive Environments.” NeurIPS (2017).
>
>
> Vojtech Kovarik, Martin Schmid, Neil Burch, Michael Bowling, and Viliam Lisy, Rethinking formal models of partially observable multiagent decision making, IJCAI 2023
>
> Johan Peralez, Aurelien Delage, Jacopo Castellini, Rafael F. Cunha, and Jilles Steeve Dibangoye, Optimally solving simultaneous-move Dec-POMDPs: The sequential central planning approach, AAAI 2025
>
> Wang, Jianhong, Wangkun Xu, Yunjie Gu, Wenbin Song, and Tim C. Green. “Multi-Agent Reinforcement Learning for Active Voltage Control on Power Distribution Networks.” NeurIPS (2021).​
>
> Wen, Muning, Jakub Grudzien Kuba, Runji Lin, Weinan Zhang, Ying Wen, Jun Wang, and Yaodong Yang. “Multi-Agent Reinforcement Learning is a Sequence Modeling Problem.” arXiv preprint arXiv:2205.14953 (2022).​
>
> Ye, Jianing, Chenghao Li, Jianhao Wang, and Chongjie Zhang. “Towards Global Optimality in Cooperative MARL with the Transformation and Distillation Framework.” arXiv preprint arXiv:2207.11143 (2022).​

---

> ### Author Response · Authors · 2025-11-27
>
> Dear Reviewer hsU2,
>
> Thank you again for taking the time to review our work. Since the initial reviews, we have updated the manuscript with:
>
> * A detailed clarification of our setting (CTDE‑DR vs CTCE/CTDE‑CR) and the relation to Ye et al. (TAD) and related work (Appendix K).​
>
> We would be very grateful if you could let us know if there are any remaining major concerns after these updates, or if the current version sufficiently addresses the earlier weakness that was raised. This would greatly help us understand how to further improve the work, and address any remaining concerns if any.

---

### Official Review · Reviewer_XG36 · 2025-11-01

**Soundness:** 3
**Presentation:** 4
**Contribution:** 4
**Rating:** 6
**Confidence:** 2

**Summary:**

The authors address cooperative Multi-Agent RL (MARL) by introducing Agent-chained belief MDPs and Agent-Chained policy iteration. Together, they contribute to the MARL field by providing a novel paradigm where a cooperative (decentralized) multi-agent problem is reformulated as a serialized decision process where agents act sequentially. The authors demonstrate that an optimal policy of such reformulation also leads to optimal behavior for the original multi-agent problem. Furthermore, the authors demonstrate a practical implemenentation of this approach achieving state-of-the-art empirical results on relevant benchmarks in the field.

**Strengths:**

- The presented method is presented via both a solid theoretical analysis and an empirical evaluation over several notorious benchmarks in MARL settings, including high-dimensional tasks.
- The paper is clearly written and relatively easy to follow.
- The presented method proves to be scalable as it's easily applicable to currently existing algorithms such as MAPPO. The authors claim that the final practical implementation leads to a similar objective to MAPPO, with the most crucial change being the sequential agent-chaining. This makes it easy to attribute the empirical performance gains primarily to the novel paradigm introduced by the authors.
- The authors also report important considerations on the runtime statistics and computational resources usage (Figure 5 and Appendix J), providing more insights on the reproducibility and scalability of their approach.

**Weaknesses:**

- The overall empirical experimental evaluation relies on only three seeds. I believe this can considerably hurt the significance of the results, and suggest the authors to increase this to at least 5 (ideally 10) to solidify the empirical findings and claims. This is also evident in Table 2 where there is considerable overlap between the standard errors, with unclear significance over performance gain by one method in particular (only 1 row displays statistically significant higher mean return for ACPO).

**Questions:**

- Does the requirement of defining a serialized decision process with sequentially acting agent restrict, in practice, the applicability of the algorithm? It's unclear to me whether there exist tasks where this can not be applied, even if considering a belief over unobservable other-agent actions. E.g. for tasks where actions must occur syncronously.
- Could the authors clarify why only 3 seeds have been used for the experimental evaluation and whether this hurts statistical significance particularly for high-dimensional tasks?
- The authors attached an anonymous URL with their codebase in Appendix I. While this would be highly beneficial to the community for reproducibility and further research, the underlying repository seems to be unavailable at the moment. Could and will the authors provide access to it?

---

> ### Author Response · Authors · 2025-11-21
>
> # Response to Reviewer XG36
>
> > **The overall empirical experimental evaluation relies on only three seeds. I believe this can considerably hurt the significance of the results, and suggest the authors to increase this to at least 5 (ideally 10) to solidify the empirical findings and claims.**
>
> We have updated Figure 2 (RWARE), Figure 3 (MA-MuJoCo), Table 2 (SMACv2) and Figure 7 (SMACv2) in our manuscript. The updated results are provided for **10 seeds** on RWARE, SMACv2, and MA-MuJoCo. There is no significant change in the empirical results, and our analysis and conclusions remain the same.
>
> We initially followed the experimental protocol in HARL (Zhong et al., 2024) which also ran 3 seeds on most tasks to ensure a wide coverage of environments, tasks and baselines, as well as statistical significance.
>
> (Note) The additional seeds for the 8 and 12 agent tasks for RWARE will be added later during the discussion phase, as training the best response baselines (HAPPO and HATRPO) requires considerably more time.
>
>
> > **Does the requirement of defining a serialized decision process with sequentially acting agent restrict, in practice, the applicability of the algorithm? It's unclear to me whether there exist tasks where this can not be applied, even if considering a belief over unobservable other-agent actions. E.g. for tasks where actions must occur syncronously.**
>
>
> Following the standard setup in cooperative MARL with Centralized Training Decentralized Execution (CTDE), we emphasize that our work specifically tackles the synchronous setting where agents act simultaneously.
>
> Serialization is a theoretical tool which provides a single-agent perspective on the underlying multi-agent problem. This perspective enables us to define decentralized value functions for each agent, and provide global optimality guarantees for ACPI. For our practical implementation ACPO, we can approximate ACPI and train decentralized policies with standard single-agent RL algorithms (e.g. PPO, SAC) on top of our agent-chaining framework. At test time, the agents take actions simultaneously.
>
>
>  ---
>
> ## References
>
> Zhong, Yifan, Jakub Grudzien Kuba, Xidong Feng, Siyi Hu, Jiaming Ji, and Yaodong Yang. “Heterogeneous-Agent Reinforcement Learning.” Journal of Machine Learning Research (JMLR) (2024).

---

> ### Author Response · Authors · 2025-11-21
>
> >**The authors attached an anonymous URL with their codebase in Appendix I. While this would be highly beneficial to the community for reproducibility and further research, the underlying repository seems to be unavailable at the moment. Could and will the authors provide access to it?**
>
> We have re-uploaded our submitted codebase to this link below: https://anonymous.4open.science/r/anonymous-acpo-E352/README.md
>
> We will provide public access to this codebase upon publication.

---

> ### Author Response · Authors · 2025-11-27
>
> Dear Reviewer XG36,
>
> Thank you again for taking the time to review our work. Since the initial reviews, we have updated the manuscript with:
>
> * Results for 10 seeds (previously 3) on RWARE, SMACv2, MA‑MuJoCo (Figure 2-3, Table 2, Figure 7) with conclusions unchanged
>
> We would be very grateful if you could let us know if there are any remaining major concerns after these updates, or if the current version sufficiently addresses the earlier weakness that was raised. This would greatly help us understand how to further improve the work, and address any remaining concerns if any.

---

### Author Response · Authors · 2025-11-21

# General Response

We thank all reviewers for taking the time to provide feedback on our paper.
We address all of the weaknesses and questions raised in the specific responses.

The following changes are made to the manuscript, which are highlighted in $\text{\textcolor{magenta}{magenta}}$.

* Additional reference to Bertsekas 2019 in Section 3 (Reviewer hsU2)
* Figure 2 of Section 7: Updated results for Multi-Robot Warehouse (RWARE) with 10 seeds (Reviewer XG36, ScUU)
* Figure 3 of Section 7: Updated results for Multi-Agent MuJoCo with 10 seeds  (Reviewer XG36, ScUU)
* Table 2 of Section 7: Updated results for SMACv2 on 10 random seeds with VDN as an additional baseline (Reviewer XG36, ScUU)
* Figure 7 of Appendix H: Updated learning curves for SMACv2 on 10 random seeds with VDN as an additional baseline (Reviewer XG36, ScUU)
* Addition of Appendix K: Details on Centralized Training Decentralized Execution (Reviewer hsU2)
* Addition of Appendix  L: Belief Approximation for Practical Implementations (Reviewer EQRP)
* Addition of Appendix M: Return vs Runtime Comparison on RWARE (Reviewer ScUU)
* Addition of Appendix N: Agent-Chained TD3 (Reviewer ScUU, EQRP)
* Additional results on Agent-Chained TD3 compared with HATD3 (Zhong et al., 2024) and MADDPG (Lowe et al., 2017) in Figure 9 of Appendix O (Reviewer ScUU, EQRP)

Please let us know if further clarifications are needed as we would be more than happy to answer them during the discussion period.

 ---

## References

Bertsekas, Dimitri P. “Multiagent Rollout Algorithms and Reinforcement Learning.” arXiv preprint arXiv:1910.00120 (2019).​

Lowe, Ryan, Yi Wu, Aviv Tamar, Jean Harb, OpenAI Pieter Abbeel, and Igor Mordatch. “Multi-Agent Actor-Critic for Mixed Cooperative-Competitive Environments.” NeurIPS (2017).

Wen, Muning, Jakub Grudzien Kuba, Runji Lin, Weinan Zhang, Ying Wen, Jun Wang, and Yaodong Yang. “Multi-Agent Reinforcement Learning is a Sequence Modeling Problem.” arXiv preprint arXiv:2205.14953 (2022).​

Ye, Jianing, Chenghao Li, Jianhao Wang, and Chongjie Zhang. “Towards Global Optimality in Cooperative MARL with the Transformation and Distillation Framework.” arXiv preprint arXiv:2207.11143 (2022).

Zhong, Yifan, Jakub Grudzien Kuba, Xidong Feng, Siyi Hu, Jiaming Ji, and Yaodong Yang. “Heterogeneous-Agent Reinforcement Learning.” Journal of Machine Learning Research (JMLR) (2024).

---

### Meta-Review · Area_Chair_96WL · 2026-01-04

**Summary:**

This paper introduces Agent-Chained Policy Optimization (ACPO), a new paradigm for multi-agent reinforcement learning (MARL) that tackles the challenge of decentralized execution. The core idea is to serialize the multi-agent decision-making process by creating an Agent-Chained Belief MDP (AC-BMDP), which effectively transforms the multi-agent problem into a single-agent problem from each agent's perspective. This allows for the application of standard single-agent RL algorithms. The authors provide theoretical grounding for their approach and demonstrate strong empirical performance on several complex MARL benchmarks, outperforming strong on-policy baselines like MAPPO.

This paper presents an interesting and creative idea, and the strong performance against MAPPO is noteworthy. However, the reviewers have also raised several concerns. The primary concern is the insufficient and incomplete experimental comparison; and the second concern is on the high computational overhead, which further limits the paper's practical impact. For a field as mature as MARL, failing to compare against a wide range of established methods in the initial submission is a major oversight. Therefore, the paper is ultimately not ready for publication at ICLR due to several unresolved concerns in its evaluation and the resulting lack of consensus.

**Reviewer Concerns:**

* *Insufficient Baselines:* The most critical and persistent weakness, raised forcefully by Reviewer ScUU, is the limited set of baseline comparisons. The initial submission only compared against on-policy methods (MAPPO/HAPPO), omitting a wide range of standard and state-of-the-art MARL algorithms (e.g., VDN, QMIX, MADDPG, COMIX). This makes it very difficult to properly situate ACPO's contribution and performance within the broader MARL landscape.

* *High Computational Cost:* The paper shows that ACPO can be 3-4 times slower than MAPPO. This is a significant practical drawback that limits the method's applicability, and the performance gains may not be sufficient to justify this high cost in all scenarios.

**Reviewer Scores:**

* Reviewer ScUU: This reviewer would maintain their score of 2. Their core concerns about baselines and computational cost were not fully resolved.

* Reviewer XG36 and EQRP: Their scores of 6 likely represent their upper bound. A full discussion highlighting the limited baselines might have even caused them to lower their scores.

* Reviewer hSU2: This review is not reliable and should be discounted.

---

### Decision · Program_Chairs · 2026-01-26

Reject